# Divergence Decoding: Inference-Time Unlearning via Auxiliary Models

Humzah Merchant [1]   Bradford Levy [1]

## Abstract

Large Language Models (LLMs) frequently memorize sensitive training data thereby creating significant privacy and copyright risks. Addressing these risks, i.e., removing such knowledge from an existing model checkpoint, has proven challenging as many unlearning methods lead to catastrophic utility loss or are ineffective for complex queries. We introduce **Divergence Decoding (DD)**, a mechanism that uses small auxiliary models to steer the logits of the LLM away from specific data during inference. Training these models is straight forward, i.e., we use standard pre-training and fine-tuning setups. We find the method decisively outperforms state-of-the-art (SOTA) baselines on unlearning benchmarks across a variety of model and training dataset scales consistent with DD being an effective and inexpensive solution to unlearning. We then demonstrate that this steered distribution can be trivially distilled back into the base model. Since the method is generally applicable to any probabilistic model, we explore its efficacy outside of text generation and find evidence of generalization to the domain of images.[1]

## 1. Introduction

Large Language Models (LLMs) are increasingly deployed in high-stakes settings such as consumer-facing APIs, legal discovery, and financial decision-making. In these applications, it is often necessary to *selectively remove* specific training information—for example, to comply with "right to be forgotten" requests, address copyright violations, prevent look-ahead bias in financial backtesting, or generally remove sensitive content (Carlini et al., 2021).

The most direct solution—retraining a model from scratch

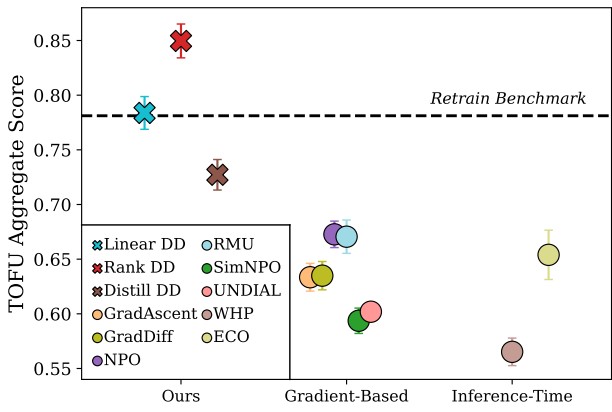

*Figure 1.* Divergence Decoding achieves near perfect performance on TOFU. 99% CI are provided. Full results in Table 5

on a sanitized dataset (Bourtoule et al., 2020)—is computationally infeasible at frontier model scales. Training a single frontier model requires millions of GPU-hours (Hoffmann et al., 2022; Grattafiori et al., 2024; DeepSeek-AI, 2025), making frequent or ad hoc unlearning requests impractical. Therefore, machine *unlearning* has emerged as a field of modifying the weights of trained models using techniques such as gradient ascent or negative preference optimization (Eldan & Russinovich, 2023; Jang et al., 2023). While cheaper than full retraining, these approaches remain expensive, fragile, and prone to degrading overall model utility via catastrophic forgetting (Kirkpatrick et al., 2017).

Our work is inspired by the Product of Experts framework (Hinton, 1999) and Importance Sampling (Hammersley & Handscomb, 1965) literature. Instead of retraining a new frontier model from scratch, we train small *auxiliary* language models using standard fine-tuning pipelines as proxies for the two frontier models. We find that *training these auxiliary models is comparatively trivial*—requiring no specialized objectives, architectural changes, or custom optimization techniques. The difference in the auxiliary models—which approximates the difference between the existing frontier model and the ideal retrained model—is used to steer generation from the frontier model.

We term this **Divergence Decoding (DD)**, and find that by adjusting the frontier model's logits using the difference between the auxiliary models, DD steers generation away

---

[1]University of Chicago, Chicago, IL 60637. Correspondence to: Bradford Levy <bll@uchicago.edu>.

*Proceedings of the $43^{rd}$ International Conference on Machine Learning*, Seoul, South Korea. PMLR 306, 2026. Copyright 2026 by the author(s).

[1]Model checkpoints and code are available on GitHub.

from unwanted content while preserving general knowledge and model utility—a desirable feature which traditional unlearning methods often fail to produce.

Given that DD requires forwarding not just the frontier model but also small auxiliary models, we conduct both theoretical analyses and extensive empirical ablations assessing the effect of our method on compute requirements and latency. We find the real-world latency impact to be minimal and proportional to the size of auxiliary models. For settings where this is still too costly, we apply a standard student–teacher distillation to transfer the behavior of a DD-augmented frontier model into a single target model to complete the unlearning process. Across TOFU and MUSE Knowledge Memorization, *we find that distillation outperforms all prior unlearning methods*.

Finally, we explore the efficacy of our method in other domains. A notable feature of our method is that it can be readily applied to any probability model. Along these lines, we train auxiliary models to target unlearning specific visual semantics in transformer-based image generation models (Esser et al., 2021). We find preliminary evidence of efficacy in image generation as well.

Our paper makes the following contributions to the literature on machine unlearning:

- a novel logit steering mechanism which exceeds current SOTA methods across standard benchmarks,

- a bridge between inference-time unlearning and weight-based unlearning via distillation,

- initial explorations of efficacy in domains beyond text,

- extensive ablations covering areas such as hyperparameter choice, compute and latency increases, cross-tokenizer and cross-family analysis, and adversarial prompting.

## 2. Related Literature

**Removing knowledge from model weights.** Model providers use methods such as Supervised Safety Finetuning and RLHF to finetune their models to reduce the likelihood of generating certain content when aligning the models (Touvron et al., 2023; Achiam et al., 2024).

**Unlearning** For post-alignment methods, a variety of different variations of finetuning (Jang et al., 2023; Zhang et al., 2024; Fan et al., 2024) or distillation (Zhong et al., 2026; Dong et al., 2025) aim to remove knowledge from the model's weights while damaging its utility as little as possible, often by using the retain set as a regulator. While these methods *can* be effective, they are generally costly and almost always result in utility loss. Other methods that edit

model weights also exist (Ilharco et al., 2023; Shen et al., 2026)

**General Inference Time Unlearning Methods** Activation-space steering computes a direction representing a conceptual contrast (e.g., "love" vs. "hate") and injects that vector during forward passes (Turner et al., 2024; Lee et al., 2025). Other methods attempt to use classifiers to classify knowledge and then disrupt the model's outputs, e.g. (Deng et al., 2025; Liu et al., 2024). We find that these tends to be binary and coarse and therefore prefer steering.

**Logit Steering Methods** Other logit steering methods have been proposed before for alignment or unlearning, all of which use a linear steering mechanism with one (Ji et al., 2024) or two full size or auxiliary models (Eldan & Russinovich, 2023; Liu et al., 2021; Huang et al., 2025; Suriyakumar et al., 2025). To our knowledge, our work is the first to propose a rank-based steering mechanism and the first to link steering back to a single model using distillation.

## 3. Method

We adopt the **Target/Retrain** terminology used in MUSE (Shi et al., 2025). The **Target** model is the model subject to unlearning: it has been trained on a dataset containing observations which should be retained and some that should be forgotten. The **Retrain** model is the gold-standard model trained from scratch only on the "retain" data. We denote the Target model by $P$ and the Retrain model by $Q$. In deployment, the Retrain model is precisely what we would like to sample from, but it is typically unavailable because retraining a frontier-scale model is prohibitively expensive.

### 3.1. Divergence Decoding

Divergence Decoding approximates the behavior of the Retrain model $Q$ while using the accessible Target model $P$ and two smaller auxiliary models. Consider two small auxiliary models $p$ and $q$. Given a prefix $x_{<t}$, each model defines a distribution over the next token $x_t$. Denote the logits of a model $M$ by $\ell_M(x_{<t}) \in \mathbb{R}^{|V|}$. Divergence Decoding approximates sampling from $Q$ by adjusting the logits of $P$ according to the difference between the auxiliary models $q$ and $p$. The auxiliary logit difference $\ell_q(x_{<t}) - \ell_p(x_{<t})$ estimates a direction that moves the Target model towards the ideal Retrain model.

Empirically, we consider two adjustments. The first is a linear combination of logits,

$$\hat{\ell}_Q^{LC}(x_{<t}) = \ell_P(x_{<t}) + \alpha \cdot [\ell_q(x_{<t}) - \ell_p(x_{<t})], \quad (1)$$

while the second adjustment is rank based,

$$\hat{\ell}_Q^{R}(x_{<t}) = \ell_P(x_{<t}) - \mathbb{1}_{\text{rank}(\ell_p(x_{<t}) - \ell_q(x_{<t})) \leq k} \cdot \infty. \quad (2)$$

Samples can then be drawn using standard decoding methods (e.g., Fan et al., 2018; Holtzman et al., 2020) from the approximation,

$$\widehat{Q}(x_t \mid x_{<t}) = \mathrm{softmax}(\hat{\ell}_Q(x_{<t})). \qquad (3)$$

### 3.2. Theoretical motivation

Our method is theoretically motivated by the Product of Experts (Hinton, 1999) and Importance Sampling (Hammersley & Handscomb, 1965) literature. In Appendix A, we show that the approximation $\widehat{Q}$ can be formulated as a Product of Experts model,

$$\widehat{Q}(x_t \mid x_{<t}) \propto \underbrace{P(x_t \mid x_{<t})}_{\text{Base Expert}} \cdot \underbrace{\left[\frac{q(x_t \mid x_{<t})}{p(x_t \mid x_{<t})}\right]^\alpha}_{\text{Domain Expert}}, \qquad (4)$$

where $\widehat{Q}$ is the product of a "Base Expert" $P$, which provides the Target model's general capabilities, and a "Domain Expert" given by the ratio between the two auxiliaries $p$ and $q$. Intuitively, the role of the domain expert can be summarized by three cases:

1. $q(x_t \mid x_{<t}) \approx p(x_t \mid x_{<t})$: The token is similarly likely under both auxiliaries, so the domain-expert ratio is close to 1 and the probabilities from the base model $P$ are largely unchanged.

2. $q(x_t \mid x_{<t}) \gg p(x_t \mid x_{<t})$: The token is much more likely under $q$ than $p$, so the domain expert upweights the token.

3. $q(x_t \mid x_{<t}) \ll p(x_t \mid x_{<t})$: The token is much less likely under $q$ than $p$, so the domain expert downweights the token.

Finally, DD can also be linked to importance sampling in Monte Carlo analysis, where the expectation of a function $f(x)$ under a target distribution is estimated using samples drawn from a proposal distribution. Formally,

$$\mathbb{E}_{x \sim D_{\text{target}}}[f(x)] = \mathbb{E}_{x \sim D_{\text{proposal}}}\left[f(x)\frac{D_{\text{target}}(x)}{D_{\text{proposal}}(x)}\right], \qquad (5)$$

where the importance weight $w(x) = D_{\text{target}}(x)/D_{\text{proposal}}(x)$ adjusts the expectation taken over the proposal distribution for differences between the proposal and target distributions. Analogously, Divergence Decoding uses the token-level ratio $q(x_t \mid x_{<t})/p(x_t \mid x_{<t})$ to adjust samples from the accessible Target model $P$ toward the behavior of the inaccessible Retrain model $Q$.

### 3.3. Model Distillation

For situations where we do not want to run three models in parallel online, we want to transfer the behavior of Divergence Decoding back into a single model's weights. We treat a **frozen Linear DD system** as the teacher and fine-tune the Target model as a student using standard teacher–student distillation (Hinton et al., 2015; Mirzadeh et al., 2019) on the **forget set**. Training is performed using only a KL-divergence objective between the student model's output and the frozen Linear DD teacher $\widehat{Q}$:

$$\mathcal{L}_{\text{unlearn}} = T^2 \cdot \mathrm{KL}\left(\sigma\left(\frac{\ell_P}{T}\right) \,\Big\|\, \sigma\left(\frac{\hat{\ell}_Q^{LC}}{T}\right)\right),$$

where $T$ is a temperature hyperparameter and $\sigma$ denotes the softmax function. Conceptually, this procedure distills the unlearning signal encoded by the auxiliary models $p$ and $q$ into the weights of the Target model, converting an inference-time controller into a single parametric network.

## 4. Experiments and Analysis

We evaluate the effectiveness of DD using the two gold-standard unlearning benchmarks: **MUSE** (Shi et al., 2025) and **TOFU** (Maini et al., 2024). All evaluations are performed using the Open Unlearning framework (Dorna et al., 2025). These benchmarks simulate an unlearning scenario whereby a model must eliminate specific knowledge (based on the 'forget' set) while preserving other capabilities (the 'retain' set).

A key difference between them is that TOFU uses instruction-tuned models to answer complex queries, and OpenUnlearning further expands this setup by penalizing incoherent generation on forget questions. In contrast, MUSE employs pre-trained models evaluated via few-shot Q&A on significantly simpler questions. The general sensitivity of instruction-tuned models to fine-tuning suggests that as a benchmark, TOFU is potentially harder than MUSE (Springer et al., 2025; Ghosh et al., 2024; Qi et al., 2025).

### 4.1. Setup

For **MUSE** (Shi et al., 2024; 2025), we use the news dataset and fine-tune *princeton-nlp/Sheared-LLaMA-1.3B* (Xia et al., 2024) for our auxiliary models. Here, $p$ is fine-tuned only on the provided forget set, while $q$ is fine-tuned on the provided retain set. For **TOFU** (Maini et al., 2024), we use the official Open-Unlearning target and retrain 1B models as our auxiliary models (Table 4). $p$ is trained on both the forget and retain sets, while $q$ is trained only on the retain set. This differs from the MUSE configuration because the TOFU retain set is roughly nine times larger than the forget set. Training $p$ only on the forget set would therefore cause $p$ and $q$ to undergo very different fine-tuning

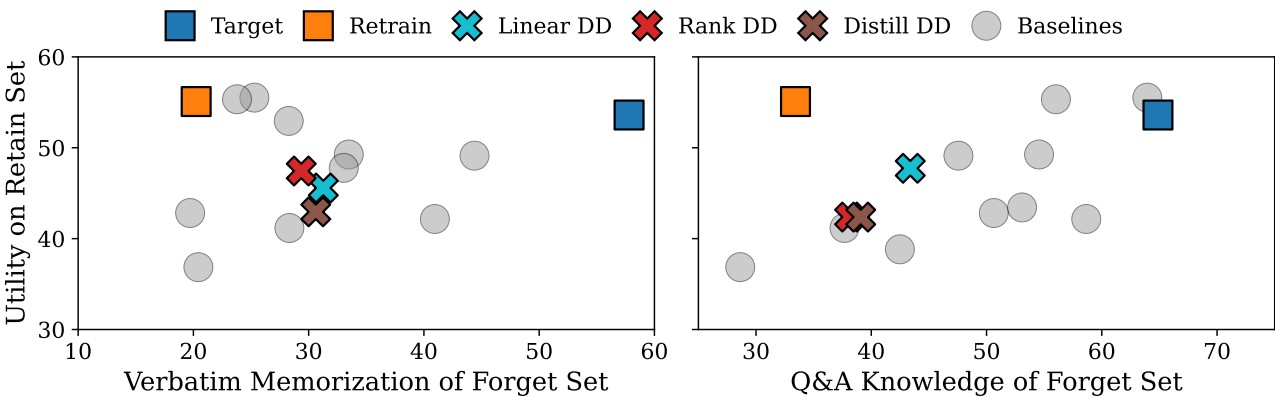

*Figure 2.* MUSE Results. Closer to retrain is better. 99% CIs are smaller than marker sizes. Detailed results split by inference-time and gradient-based methods are available in Figures 14 and 15.

shifts, making the contrast between them reflect dataset size and fine-tuning dynamics rather than the effect of removing the forget set.

To assess the trade-off between utility and forgetting across hyper-parameter choices, algorithm variants, and model sizes, we employ distinct metrics suited to each benchmark's complexity. For **MUSE**, we report the Euclidean distance to the Retrain baseline, normalized such that the Target model represents 100%. This provides an all-encompassing score capturing both Q&A and memorization performance. For **TOFU**, we utilize the aggregate score (with and without privacy), as detailed in Appendix F of Dorna et al. (2025).

### 4.2. Primary Result

Figures 1 and 2 present the core result of the paper. Divergence Decoding substantially outperforms prior methods on TOFU, marginally improves over the strongest baselines on MUSE Knowledge Memorization, and is less effective on MUSE Verbatim Memorization. This pattern roughly follows task complexity: TOFU uses instruction-tuned models to answer longer, more complex queries; MUSE Knowledge Memorization relies on shorter few-shot questions; and MUSE Verbatim Memorization is lifted directly from the training text, where simpler filtering or classification-style methods can be especially strong. We hypothesize that a key advantage of Divergence Decoding is that it leverages the semantic and reasoning capabilities of the auxiliary models, which becomes most useful when forgetting requires more than suppressing near-verbatim strings.

### 4.3. Algorithm and Hyper-Parameter Choice

Figure 3 studies Linear DD and Rank DD on both benchmarks across a wide sweep of hyper-parameters. On MUSE, we find that Rank DD outperforms on memorization while Linear DD marginally outperforms on Q&A. On TOFU,

Rank DD marginally outperforms Linear DD. We find that there is a large range of hyper-parameter values that work well for both. Additional data in Figure 11 and Table 7.

### 4.4. Model Size

Given that our method works well with the 1B and 1.3B small models, a natural question is how sensitive performance is to the size of $p$ and $q$. We investigate this using the 2.7B Sheared-LLaMA model variant for MUSE, the Llama 3.2 3B variants for TOFU models, and trigram LMs based on *Stupid Backoff* (Brants et al., 2007) for both. Our trigram implementation pre-computes all scores as arrays the size of the vocabulary, effectively giving zero inference overhead and serving as a limiting case of "0% of $P$'s size."

We evaluate the most optimal configuration for each model size. On MUSE, scaling from 1.3B to 2.7B yields a noticeably larger gain than the corresponding jump from 1B to 3B on TOFU. Meanwhile, the trigram models—which perform surprisingly well on some MUSE settings—fail almost entirely on TOFU. Upon further inspection of the Q&A questions on MUSE where the Trigram models perform well, we find that this is largely due to questions which are more similar to the underlying training data. Thus, we conclude that the Trigram models are likely most useful for unlearning verbatim content.

### 4.5. Privacy, Over-Unlearning, and Calibration

The naive implementation of the *rank based method*—e.g., setting targeted logits to $-\infty$—would produce degenerate privacy scores, since the losses would be infinite. To preserve the ability to evaluate over- versus under-unlearning in the rank-based setting, we instead replace the $k$ most divergent logits with the $k$th largest logit in the unmodified distribution. Across both MUSE and TOFU, a broad range of hyper-parameters produce models that are statistically **in-**

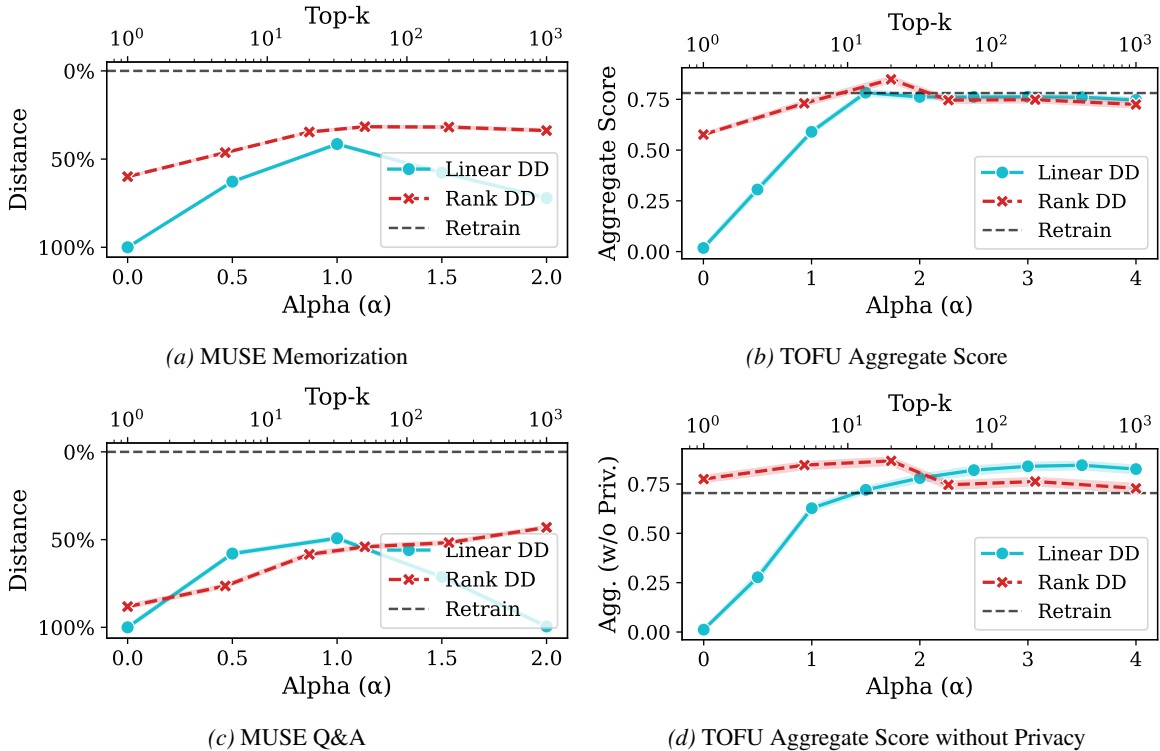

*(a)* MUSE Memorization

*(b)* TOFU Aggregate Score

*(c)* MUSE Q&A

*(d)* TOFU Aggregate Score without Privacy

*Figure 3.* Effect of hyper-parameter and algorithm choice. 99% CI are provided.

**distinguishable** from a full retrain, striking a clean balance between over- and under-unlearning. The fact that the optimal region occurs around $\alpha > 1$ aligns with the intuition from §3.1 that a simple linear combination of logits may be a near-optimal solution.

**Over-unlearning, even when utility is preserved, is not always optimal.** In settings like toxic content prevention, aggressively suppressing certain outputs is reasonable. However, many real-world applications are highly sensitive to *over*-unlearning. For instance, in financial modeling—such as backtesting trading strategies or stress testing banks—the goal is to evaluate performance using only the information that would have been available at the time. For example, one would want to unlearn the 2008 financial crisis so they could realistically assess the performance of an LLM making decisions at the time. **Over-unlearning would cause the model to overcompensate** to the point that it assigns even lower likelihoods to events than what was expected at the time (e.g., if the true expectation at the time of a recession were 20% an over-unlearned model would assign less than that.)

### 4.6. Distillation

Firstly, we study the effect of the distillation on the model to understand what layers it targets. We froze the target model and computed the DD distillation loss (output KL), then backpropagated without updating weights. We measured

the $L_2$ norm of gradients on the MLP and attention weights at each layer. As shown in Figure 6, we find that the gradient propagates throughout the network and peaks in intermediate layers where prior work suggests factual associations are often stored (Meng et al., 2022), and the distill is not simply a bandaid on the final layers of the model.

Across benchmarks, the distilled model substantially outperforms prior gradient-based approaches, recovering most of the improvement achieved by Linear DD. Figure 7 demonstrates a smooth surface across the temperature and learning rates grid for TOFU. We also present Figure 13 in the appendix, though we find the surface to be less smooth and it is harder to find a global optimum.

### 4.7. Cross Tokenizer

We test whether DD depends on auxiliaries from the same family. To address this, we evaluated *allenai/OLMo-2-0425-1B(-Instruct)*, *google/gemma-3-1b-(pt/it)*, and *Qwen/Qwen3-1.7B(-Base)*. We implemented a simple cross-tokenizer bridge. At initialization, each auxiliary token is decoded to text and re-encoded with the main tokenizer to find exact matches; if no exact match exists, we progressively shorten prefixes until a match is found. For example, if the auxiliary has "abcd" and the main model contains "abcd", "abcde", and "abcdef", we map the auxiliary token's logit difference onto all compatible main-model tokens. This mapping is

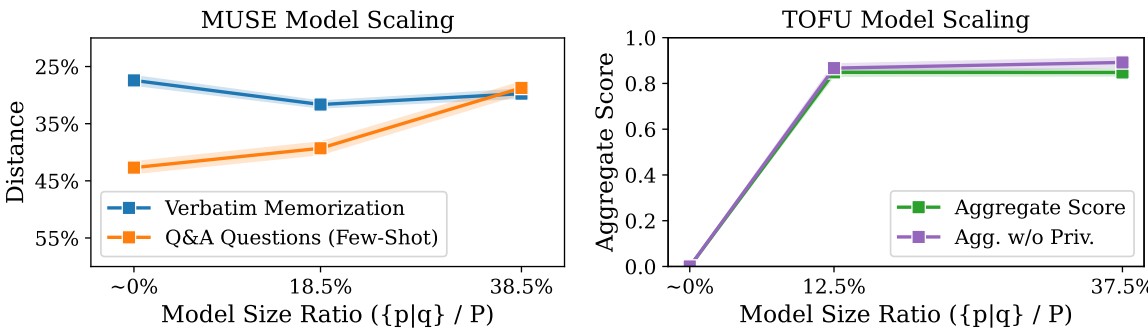

*Figure 4.* Analysis of model scaling on MUSE and TOFU. 99% CI are provided.

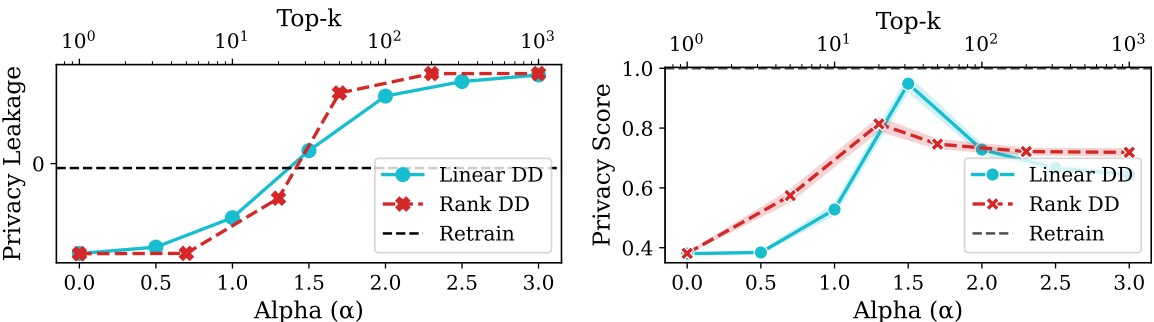

*Figure 5.* Analysis of Over- or Under- Unlearning on MUSE (left) and TOFU (right). Closer to retrain is better. The optimal values for both benchmarks are Alpha~1.5 and TopK~20. 99% CI are provided.

built once and then applied at inference with a single vectorized scatter, so the additional overhead is negligible. In addition, we are careful when handling system prompts and special tokens.

As shown in Figure 16 and Table 6, we find that these cross-family auxiliaries remain close to the same-family baseline and still competitive with the broader set of unlearning methods.

### 4.8. Adversarial Prompting

A natural concern is whether the apparent unlearning is robust to prompt variation and repeated sampling under probabilistic decoding. To evaluate this, we apply the Leak@K benchmark (Reisizadeh et al., 2025) on TOFU, which measures whether forgotten information can be recovered across multiple generation attempts. Figure 17 shows that **Distill DD matches the Retrain model**, while Linear DD and Rank DD also remain competitive. These results suggest that Distill DD does more than apply a brittle surface-level correction: the forgotten information remains difficult to recover even under repeated adversarial prompting.

### 4.9. Sustainability and Scaling

Finally, prior work has found that many unlearning methods exhibit poor scalability—the unlearning of very large amounts of content—and sustainability—sequential requests to unlearn additional content. We explore the efficacy of our method along these dimensions using the MUSE scaling and sustainability benchmarks to ensure that performance does not degrade. To extend the benchmark, we use our aggregate score which includes performance on the original forget set (Q&A), ensuring that larger and subsequent forget requests **do not** come at the cost of the forget weights being overwritten. We also evaluate sequential distillation to ensure it does not result in catastrophic forgetting. As demonstrated by Figure 12, both the inference-time and distill DD perform well, with all methods except GradDiff within the margin of error of eachother.

### 4.10. Cost and Latency Analysis

A key consideration of applying our method is the increased inference-time compute from running the two small models in tandem with the large model. Denote the number of parameters in the large model as $N$ and $n$ the number in each small model. Following the approximation of Kaplan et al. (2020), the total inference cost increases from $2N \longrightarrow 2(N + 2n)$, with the relative increase given by

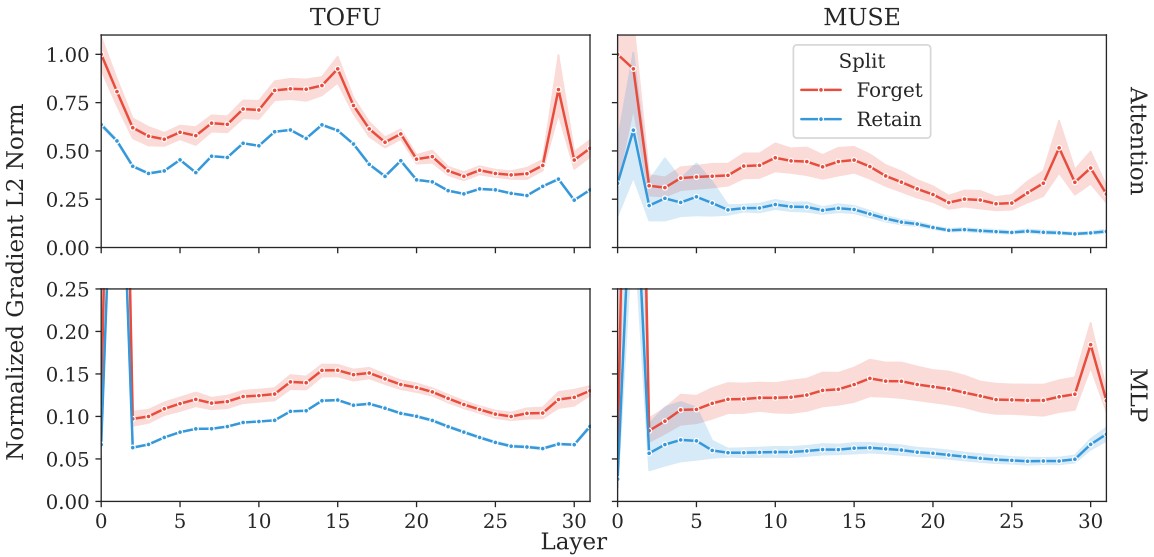

*Figure 6.* DD distillation $L_2$ norm of gradients at each layer. 99% CI are provided

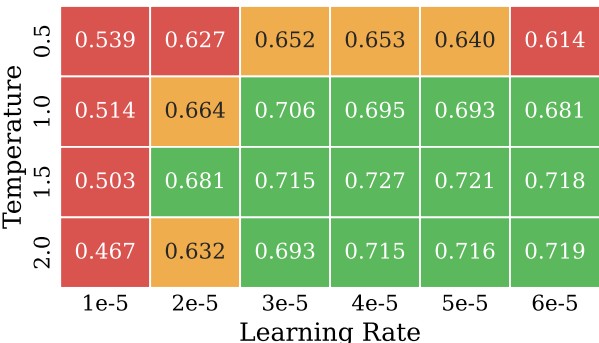

*Figure 7.* Aggregate scores for distilling setups on TOFU. The majority of hyper-parameter choices (green) outperform the SOTA. Epochs are fixed to 10 and $\alpha = 1.5$ for the DD

$2n/N$. In Figure 8, we empirically measure the increase in compute costs associated with running over 1,200 different combinations of models in a distributed setting within a single 8x$\{H100|B200\}$ instance (see Appendix B for details). We find that the compute costs tend to scale closely with our theoretical approximation. In Appendix B, we examine the effect of our method on latency and find that the increase is generally less than 0.1%.

## 5. Beyond Text

One benefit of our method is its generality, i.e., it can be applied to any setting where samples are drawn from some distribution $P$ and data exists to estimate $p$ and $q$. Along these lines, we explore the extent to which our method is effective in domains beyond text by applying it to image generation.

We begin with the setup of Esser et al. (2021) and augment the sampling in latent space per equations 1 and 2. The models $p$ and $q$ are estimated using data from the train split of ImageNet associated with the dog synset. Specifically, half the descendants from the dog synset are randomly assigned to the forget set $F$ and the other half to the retain set $R$—notably, this random assignment ensures that any preferences over dog classes will be uncorrelated with the assignment to retain versus forget. The class-conditional ImageNet checkpoint from Esser et al. (2021) is then fine-tuned on $F$ and $R$ to estimate $p$ and $q$, respectively. We then sample images from the model configured without any divergence decoding (Baseline) and with various linear and rank-based DD configurations. (Figure 9 and Appendix D).

As a first quantitative evaluation of the efficacy of our method, we evaluate the content of class conditional generations using VQAScore (Lin et al., 2024). For each class conditional generated image, we prompt a multi-modal LLM (MLLM) to assess whether the image contains the specific class and take the probability of "Yes" as the VQAScore— we rely on GPT-4o-mini for this task as it requires access to the log probabilities and many modern closed models do not provide this, e.g., GPT-5-nano. Table 1 presents mean VQAScores for class conditional samples split by whether the class was assigned to the retain or forget set. For linear divergence decoding setups, modest settings of alpha display efficacy, e.g., $\alpha = 1$ decreases the mean VQAScore on classes in the forget set from 97% to 20%. This is similar to our findings within the text domain where $\alpha$ in the range of 1 to 2 typically yielded the best results. In contrast, the rank-based setups require larger values for top-k to reach similar efficacy, e.g., $topk = 250$.

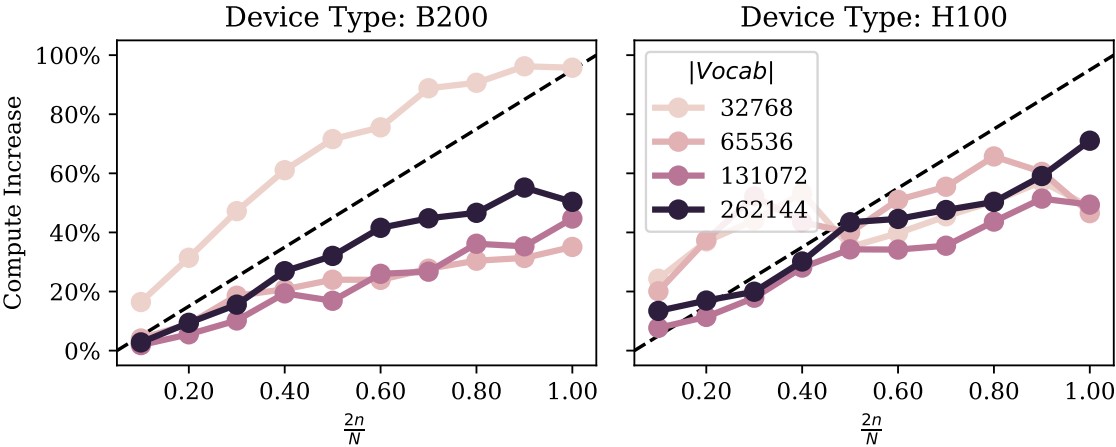

*Figure 8.* Empirical increases in compute requirements for a sample of more than 1,200 models. Size of $P$ ranges from 300M to 80B.

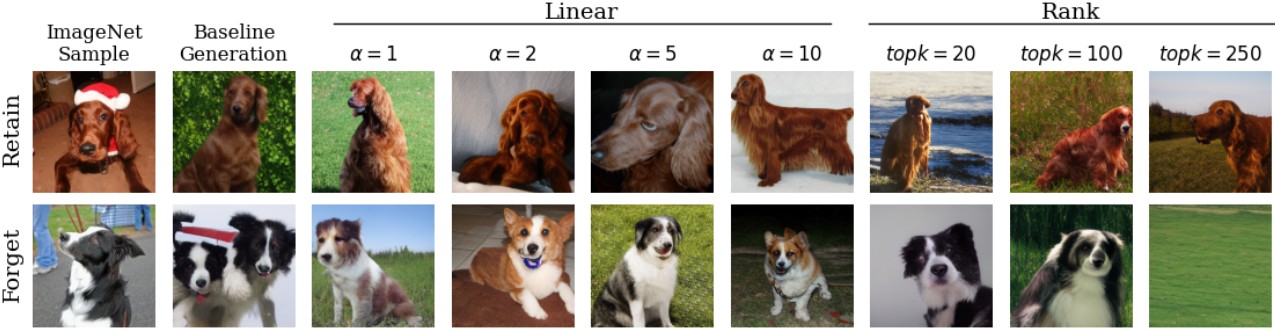

*Figure 9.* ImageNet examples, baseline generations, and generations under various divergence decoding setups.

As a second evaluation of the efficacy of our method, we evaluate the perceptual quality of generated images. Notably, a naive unlearning method could simply output noise for classes in the forget set. While this would constitute "unlearning," it may not be particularly useful if the desired outcome is perceptually similar and plausible generations without the *indicia* associated with the classes to be forgotten, e.g., the identifiable style attributable to an artist requesting that a model provider adhere to copyright laws. Along these lines, we follow Chen et al. (2024) and employ an MLLM-as-a-judge to perform pairwise comparison of the visual quality between samples from our baseline setup and a given divergence decoding setup.

Tables 2 and 8 present the performance for a variety of divergence decoding setups. In general, there is little decrease in the perceptual quality on samples conditional on classes in the retain set. For those in the forget set, however, there is a decrease in quality. For example, a setup with $\alpha = 5$ decreases the rate at which a generated image contains the class to be forgotten from 97% to 1%, but these images are also only preferred over baseline generations 31% of the time. As such, in practice one would have to sample, on

average, two generations to get a sample which both does not contain the class to be forgotten and meets or exceeds the baseline quality.

*Table 1.* Content analysis of images generated using various divergence decoding setups. Mean ± standard error. GPT-4o-mini is used as a judge.

| Method | Config | Retain | Forget |
|---|---|---|---|
| Baseline | — | $96\% \pm 1.1$ | $97\% \pm 1.0$ |
| Linear | $\alpha = 1$ | $96\% \pm 1.2$ | $20\% \pm 2.4$ |
| Linear | $\alpha = 2$ | $97\% \pm 1.0$ | $20\% \pm 2.3$ |
| Linear | $\alpha = 5$ | $96\% \pm 1.1$ | $1\% \pm 0.6$ |
| Linear | $\alpha = 10$ | $96\% \pm 1.2$ | $1\% \pm 0.6$ |
| Rank | $topk = 20$ | $96\% \pm 1.1$ | $77\% \pm 2.5$ |
| Rank | $topk = 100$ | $95\% \pm 1.3$ | $58\% \pm 2.9$ |
| Rank | $topk = 250$ | $95\% \pm 1.2$ | $20\% \pm 2.4$ |

## 6. Is Steering Sufficient?

While traditional unlearning has often focused on modifying a model's parameters—and indeed ablations involving distil-

Table 2. Perceptual quality analysis under Gemini 2.5 Flash-Lite. A variety of MLLM judges are presented in Table 8. Mean ± standard error.

| Method | Config | Retain | Forget |
|--------|--------|--------|--------|
| Linear | $\alpha = 1$ | $52\% \pm 1.7$ | $38\% \pm 1.4$ |
| Linear | $\alpha = 2$ | $49\% \pm 1.6$ | $37\% \pm 1.3$ |
| Linear | $\alpha = 5$ | $52\% \pm 1.6$ | $31\% \pm 1.2$ |
| Linear | $\alpha = 10$ | $47\% \pm 1.6$ | $31\% \pm 1.3$ |
| Rank | $topk = 20$ | $49\% \pm 1.6$ | $47\% \pm 1.6$ |
| Rank | $topk = 100$ | $47\% \pm 1.6$ | $50\% \pm 1.5$ |
| Rank | $topk = 250$ | $48\% \pm 1.6$ | $20\% \pm 1.1$ |

lation indicate this is effective—our main results involving DD suggest that this need not be the case. Specifically, the MUSE and TOFU results suggest that in many practical use cases modifying the model weights is not only unnecessary, it is comparatively harmful to model utility:

**Compliance with "Right to be forgotten" and addressing copyright violations**. These requests are numerous and ad hoc which poses a challenge for traditional unlearning methods as highlighted in Figure 12. In a similar way, there are numerous examples of alleged copyright infringement by model providers. Our method, specifically the **n-gram based setup**, excel at eliminating verbatim memorization tasks (see Figure 4) with **virtually no cost.** We envision a scenario where our method can be used to quickly eliminate such violations, particularly the most egregious cases where copyrighted material is regurgitated verbatim.

**Guardrails on API locked models** For example, intentionally training $p$ on a large amount of toxic content and $q$ on the same data distribution as $P$ could be used to approximate a $\hat{Q}$ with reduced toxic content generation. As we discuss in Section 4.10 and Appendix B, in **production-like environments** compute costs tend to scale close to theoretical approximation, and there is virtually no increase in latency.

**Low-Volume Deployments (e.g., Financial Backtesting)** As detailed in Appendix B.3, there is a computational trade-off between inference-time steering and distilling a final model. While distillation reduces inference latency, it incurs a heavy upfront training cost. If the total expected inference volume ($I$) is low—such as in one-off backtesting runs or targeted analysis—it is more FLOP-efficient to simply steer the model using Divergence Decoding rather than investing compute to distill a standalone unlearned model.

## 7. Conclusion

Traditional approaches to machine unlearning have treated the problem as one of "forced forgetting"—modifying a

large model's weights directly, often at the cost of model utility. Our findings present an alternative to this paradigm by demonstrating that unlearning can be effectively solved through steering the generation of a target model based on the difference between two auxiliary models. By training small auxiliary models to learn specific distributions (forget and retain sets), we achieve precise control over model outputs that surpasses current **state-of-the-art (SOTA)** benchmarks. Further, distilling a frozen divergence-decoding setup into a single set of model weights yields a model checkpoint that can be served without any latency consequences. This method, validated across both text and image domains, suggests that the path to safer, compliant AI lies not in crippling the base model, but in guiding it. As legal and privacy requirements for LLMs continue to tighten, our "learning over forgetting" approach provides a practical path forward that redefines SOTA performance for high-stakes deployments.

## Acknowledgments

We acknowledge generous financial support from the Booth School of Business and the Center for Applied AI. This research was supported in part by the Pythia computing cluster at The University of Chicago Booth School of Business which is funded by the Office of the Dean.

We thank Ralph Koijen, Sanjog Misra, and Ted Sumers for helpful discussions. An earlier version and subset of the analyses in this paper (Merchant & Levy, 2026) appeared in the NeurIPS 2025 Workshop: Generative AI in Finance. We thank the reviewers and session participants for valuable feedback.

## Conflict of Interest Disclosure

The authors have no competing interests to disclose.

## Impact Statement

In general, we intend unlearning to support beneficial use cases - for debiasing models, preventing toxic and copyrighted content generation, and legitimate research in other domains such as finance. However, we acknowledge the approach could be misused to induce undesirable or harmful biases.

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

## A. Connection to Product of Experts

Hinton (1999) introduced the Product of Experts (PoE) framework whereby $n$ probability models are multiplicatively combined into a single model. Let the $i$-th expert be denoted by $f_i(x|\theta_i)$, then a PoE model $R$ comprised of $n$ experts is given by,

$$R(x|\theta_1, ..., \theta_n) = \frac{1}{Z} \prod_{i=1}^{n} f_i(x|\theta_i), \tag{6}$$

where $Z$ is a normalization constant. To highlight the connection between divergence decoding and PoE, recall Eq. 1:

$$\hat{l}_Q(x_{<t}) = l_P(x_{<t}) + \alpha \cdot [l_q(x_{<t}) - l_p(x_{<t})].$$

In Eq. 1, a given model $M$ has logits which are equal to the log-probabilities up to an additive constant which depends on the token sequence prefix $x_{<t}$ but not the token $x_t$, i.e.,

$$l_M(x_{<t}) = \log M(x_t|x_{<t}) + C_M(x_{<t}). \tag{7}$$

Substituting Eq. 7 into Eq. 1 for each model, gathering the constants, and performing some algebra reveals the link to PoE:

$$\log \widehat{Q}(x_t|x_{<t}) = \log P(x_t|x_{<t}) + \alpha \cdot [\log q(x_t|x_{<t}) - \log p(x_t|x_{<t})] + C$$
$$\widehat{Q}(x_t|x_{<t}) \propto \exp \big( \log P(x_t|x_{<t}) + \alpha \cdot [\log q(x_t|x_{<t}) - \log p(x_t|x_{<t})] \big)$$
$$\propto P(x_t|x_{<t}) \cdot q(x_t|x_{<t})^\alpha \cdot p(x_t|x_{<t})]^{-\alpha}$$
$$\propto P(x_t|x_{<t}) \cdot \left[ \frac{q(x_t|x_{<t})}{p(x_t|x_{<t})} \right]^\alpha.$$

## B. Detailed Analysis of Compute and Runtime Costs

In this section we explore the compute and runtime costs associated with our method using theoretical and empirical analyses. Additionally, we compare these costs to those associated with other unlearning methods to provide guidance on when our method is desirable. In general, we find that our method introduces minimal latency (less than 0.1% increases in realistic production environments) and compares favorably to other unlearning methods for a wide range of compute budgets.

### B.1. Compute Requirements

As presented in Section 4.10, the compute increase associated with our method can be approximated as $2n/N$ where $n$ and $N$ are the number of parameters in the small and big models, respectively. For example, applying our method to a 10B parameter model using 1B parameter small models is expected to require a 20% increase in compute. While this is a useful theoretical approximation, we empirically explore this approximation using a distributed setup on 8xH100 and 8xB200 instances using more than 1,200 unique combinations of models for $P$, $p$, and $q$.

For our specific setup, we target an environment where the small models $p$ and $q$ are running on some number of accelerators while multiple copies of the large model $P$ is running on additional accelerators. We consider the set of candidate models for $P$, $p$, and $q$ as those listed in Table A9 of Hoffmann et al. (2022) and add several models in the range of 19-70B parameters following the Llama 3 architecture (Grattafiori et al., 2024). Additionally, we consider four vocabulary sizes for each model: $2^{15}$, $2^{16}$, $2^{17}$, and $2^{18}$.

We then match models for $p$ and $q$ to $P$ such that $2n \leq N$ and only consider models for $P$ where $N > 8e9$. The compute increase required to run a given combination of models is then measured as the ratio $(t_P + t_p + t_q)/t_P$ where $t$ is the time required to run the models measured in GPU-hrs. Results for all combinations of models, vocabulary sizes, and devices are presented in Figure 8. In general, we find a strong agreement with the theoretical approximation.

### B.2. Latency

An additional consideration of applying our method is latency, i.e., many applications require fast responses to users and the an increase in latency of 10-20% could be unacceptable. Following from Section B.1 above, we explore the latency impact

of our method in a distributed environment where the small models $p$ and $q$ can be run in parallel with multiple copies of $P$. In this setting, the primary contributor to latency is the time required to sync the logits in Eq. 1 across devices such that sampling from the approximation to $Q$ can be performed.

Along these lines, we measure the increase in runtime as the ratio $t_Q/t_P$ where the time $t$ is the total time required to generate a sequence of fixed length, $t_Q$ is the time to do this under the distributed divergence decoding setup, and $t_P$ is the time to do this under a setup where $P$ is run on a single GPU with no synchronization overhead. The two key factors here are the vocabulary size which determines the size of the data being synchronized across GPUs and the size $N$ of $P$ which determines the baseline runtime required. Figure 10 shows that for most model configurations, the increase in runtime is less than 0.1%. For smaller "large" models, i.e., $N < 20e9$, and the largest vocabulary size, the increase is in runtime is roughly 0.2-0.5%. Thus, while there is undeniably an increase in latency, it is relatively modest at $< 0.5\%$ for the vast majority of realistic model configurations.

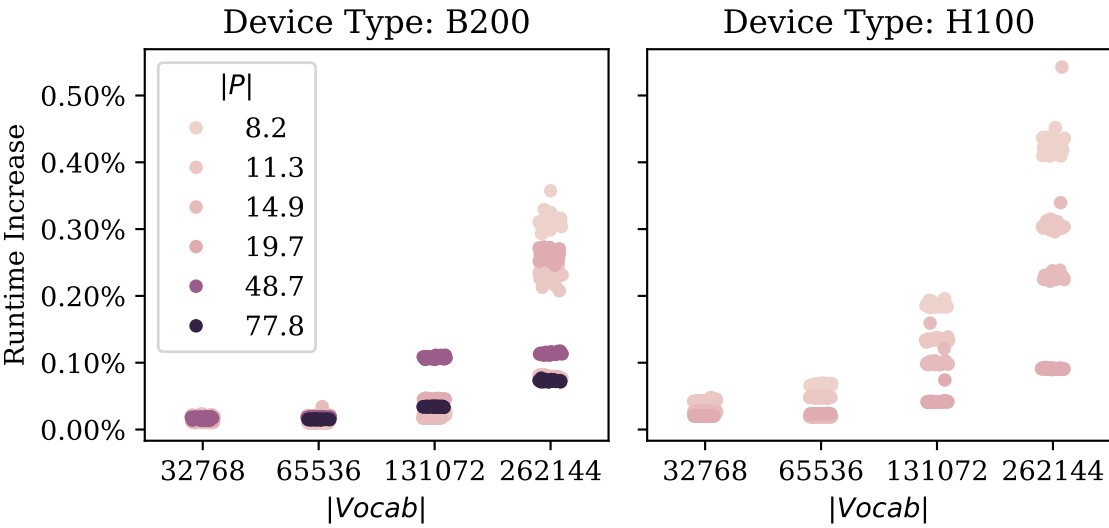

*Figure 10.* Effect of model and vocabulary size on runtime for two generations of accelerators

### B.3. Cost of Steering vs Distilling

We compare the computational cost of using Divergence Decoding (DD) directly at inference time versus using DD to distill a single unlearned model. Let $d_{retain}$ and $d_{forget}$ be the dataset sizes (in tokens), $N$ and $n$ be the parameters of the large and small models, and $e_N$ and $e_n$ be the number of training epochs. Since both methods require training the auxiliary models (the "teacher"), that cost appears on both sides of the inequality. Therefore, steering becomes more costly than distilling only when the extra inference compute of the auxiliary models exceeds the cost of training the student model. Assuming the student is trained on the forget set, this occurs when:

$$\underbrace{6ne_n(d_{\text{retain}} + d_{\text{forget}})}_{\text{Train Aux}} + \underbrace{2(N + 2n)I}_{\text{Steering Inf.}} \geq \underbrace{6ne_n(d_{\text{retain}} + d_{\text{forget}})}_{\text{Train Aux}} + \underbrace{(6N + 4n)\, e_N d_{\text{forget}}}_{\text{Distill Train}} + \underbrace{2NI}_{\text{Student Inf.}}$$

$$2(N + 2n)I \geq (6N + 4n)e_N d_{\text{forget}} + 2NI$$

$$2NI + 4nI \geq (6N + 4n)e_N d_{\text{forget}} + 2NI$$

$$4nI \geq (6N + 4n)e_N d_{\text{forget}}$$

$$I \geq \frac{(6N + 4n)e_N d_{\text{forget}}}{4n}$$

$$I \geq \frac{3Ne_N d_{\text{forget}}}{2n} + e_N d_{\text{forget}}.$$

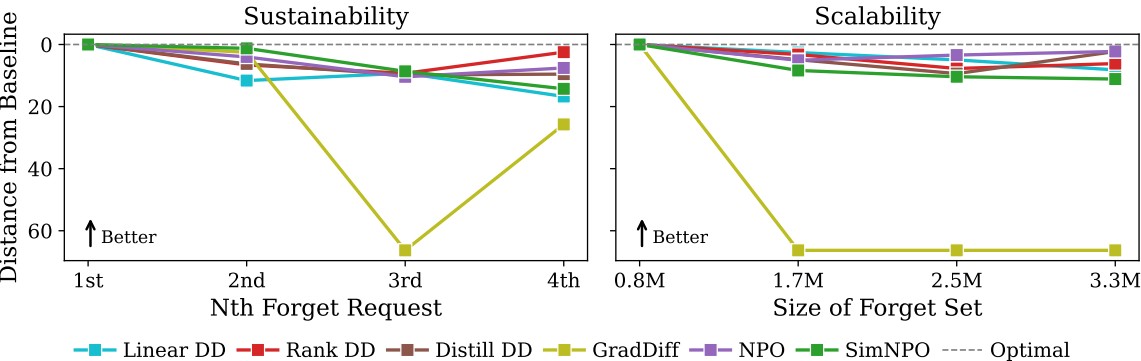

*Figure 12.* The left column is sustainability - consecutive forget sets of the same size - and the right column is scaling, increasingly large forget sets. We consider euclidean distance to the method's baseline performance when evaluated on the retain set and the **original** forget set, with the increasing distance capturing both **utility loss** and **loss of forgetting.** In general, all methods except for GradDiff perform reasonably well and within the margin of error of each other.

## C. Detailed Experimental Setups

### C.1. MUSE

We finetune the LlaMA models using the **AdamW Torch optimizer** and a **cosine scheduler** for **10** epochs. We set the learning rate such that the loss approximately halves over the course of training. We swept the LLaMA models with $\alpha \in \{0.5, 0.6, \ldots, 1.5\}$ and top-$k \in \{1, 5, 20, 50, 100, 200, 500, 1000\}$ and the trigram models at $\alpha \in \{5, 10, \ldots 30\}$ and top-$k \in \{1, 2, 3, 5, 10\}$.

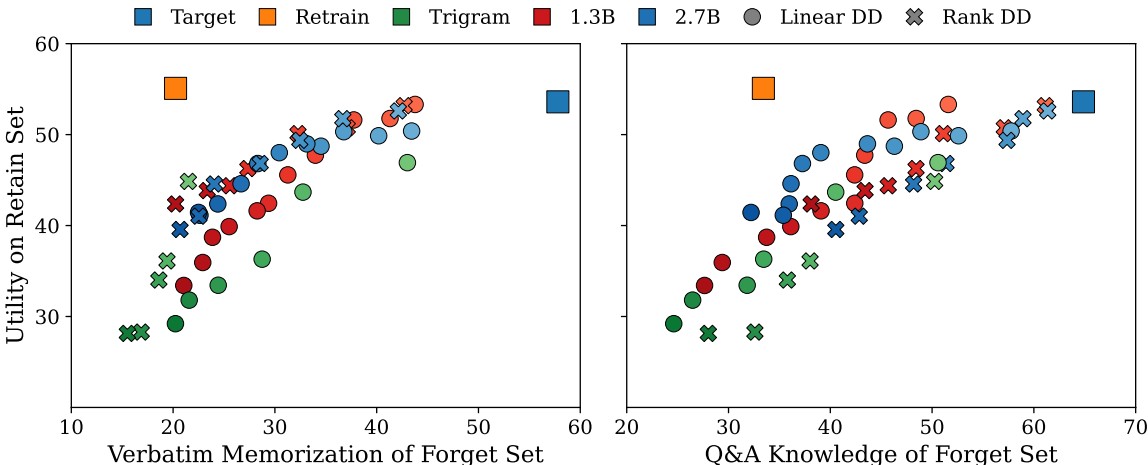

*Figure 11.* All hyper-parameter and model size configurations. Increasing values are darker and usually to the bottom and left.

*Table 3.* Configuration MUSE

| Model | Initial LR | Best Verbatim | Best Q&A |
|---|---|---|---|
| Stupid Backoff Trigram | | TopK=1 | Alpha=10 |
| princeton-nlp/Sheared-LLaMA-1.3B | 5e-5 | TopK=100 | Alpha=0.8 |
| princeton-nlp/Sheared-LLaMA-2.7B | 4e-5 | TopK=200 | Alpha=1.0 |

For the other methods, we use the default settings provided by OpenUnlearning.

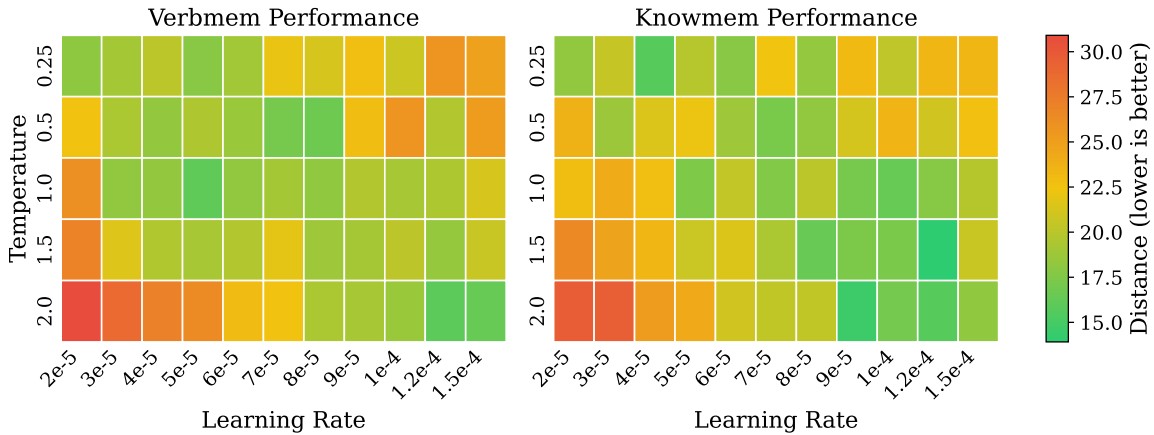

*Figure 13.* Epochs are fixed at 5. $\alpha = 0.85$, the average of the optimal for Knowmem (0.8) and Verbmem (0.9). The surface is less smooth than TOFU

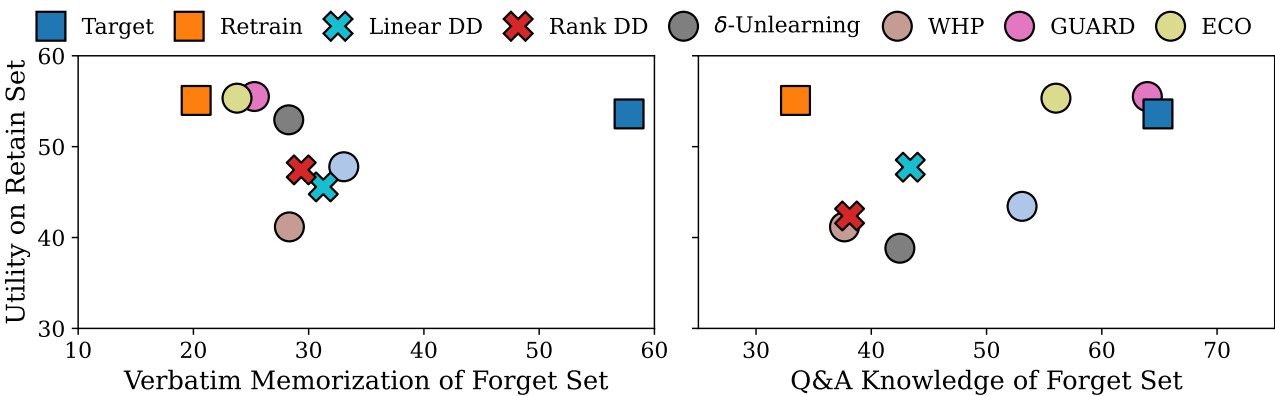

*Figure 14.* MUSE results for inference-time methods. Closer to Retrain is better.

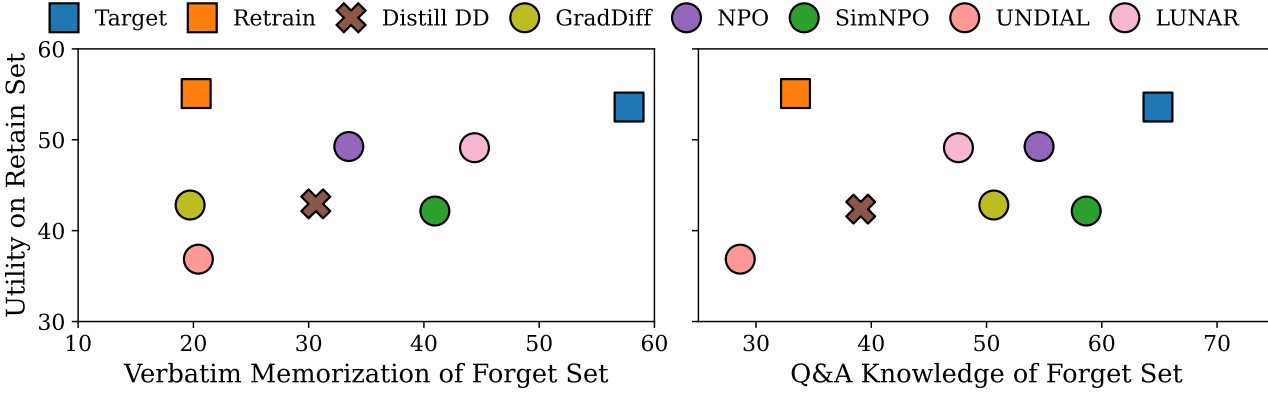

*Figure 15.* MUSE results for gradient-based methods. Closer to Retrain is better.

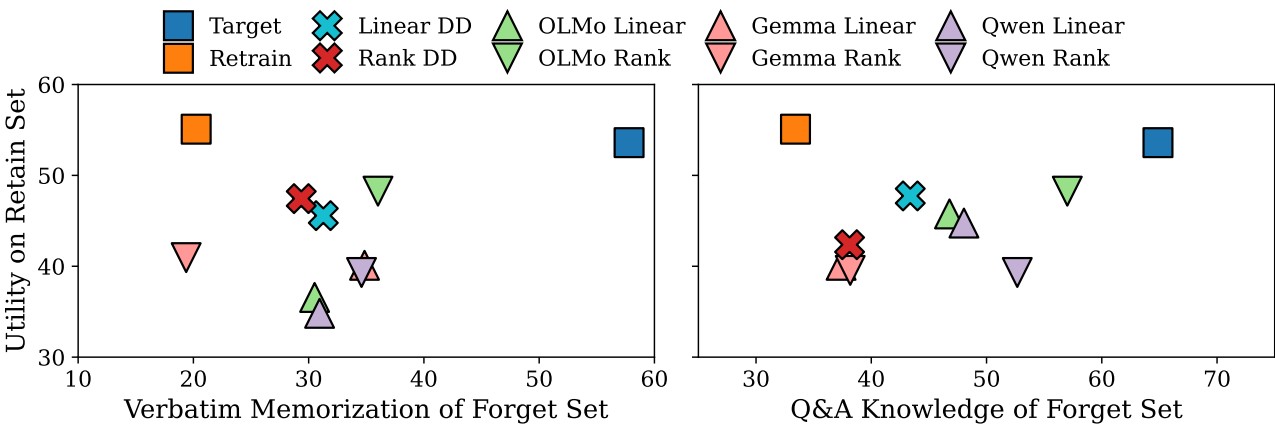

*Figure 16.* MUSE results for cross tokenizer. Closer to Retrain is better.

## C.2. TOFU

| Role | Model |
|------|-------|
| $p$ | LLaMA-3.2-1B-IT (full) |
| $P$ | LLaMA-3.1-8B-IT (full) |
| $q$ | LLaMA-3.2-1B-IT (retain90) |
| Benchmark ($Q$) | LLaMA-3.1-8B-IT (retain90) |

*Table 4.* TOFU Models

For $p$ and $q$ we use *open-unlearning/tofu_Llama-3.2-1B-Instruct_full*, *open-unlearning/tofu_Llama-3.2-1B-Instruct_retain90*, and the counterparts for 3B. For finetuning and distillation based methods, we grid search learning rates $\{5 \times 10^{-7}, 8 \times 10^{-7}, 1 \times 10^{-6}, 2 \times 10^{-6}, 3 \times 10^{-6}, 4 \times 10^{-6}, 5 \times 10^{-6}, 1 \times 10^{-5}\}$ and epochs from 1 to 10. For other methods, such as WHP, GUARD, and ECO, we grid sweep relevant hyper-parameters within a reasonable range.

*Table 5.* TOFU Results

| Method | Config | Agg. ↑ | Mem. ↑ | Priv. ↑ | Utility ↑ |
|---|---|---|---|---|---|
| Target | Full | $0.02 \pm 0.01$ | $0.01 \pm 0.00$ | $0.38 \pm 0.00$ | $1.00 \pm 0.04$ |
| Retrain | Retain90 | $0.78 \pm 0.01$ | $0.53 \pm 0.02$ | $0.98 \pm 0.01$ | $1.03 \pm 0.04$ |
| Linear DD | $\alpha$=1.5 | $\mathbf{0.78 \pm 0.01}$ | $0.56 \pm 0.02$ | $\mathbf{0.95 \pm 0.03}$ | $\mathbf{1.00 \pm 0.04}$ |
| Rank DD | k=20 | $\mathbf{0.85 \pm 0.02}$ | $0.80 \pm 0.01$ | $0.81 \pm 0.02$ | $0.95 \pm 0.05$ |
| Distill DD | lr=4e-05, T=1.5 | $0.73 \pm 0.01$ | $0.51 \pm 0.02$ | $\mathbf{0.89 \pm 0.03}$ | $0.95 \pm 0.03$ |
| GradAscent | lr=2e-6, e=3 | $0.63 \pm 0.01$ | $0.51 \pm 0.01$ | $0.61 \pm 0.02$ | $0.87 \pm 0.04$ |
| GradDiff/GA-GDR | lr=2e-6, e=3 | $0.64 \pm 0.01$ | $0.52 \pm 0.01$ | $0.62 \pm 0.02$ | $0.86 \pm 0.04$ |
| NPO | lr=4e-6, e=2 | $0.67 \pm 0.01$ | $0.57 \pm 0.01$ | $0.68 \pm 0.02$ | $0.82 \pm 0.03$ |
| RMU | lr=8e-7, e=4 | $0.67 \pm 0.02$ | $0.60 \pm 0.01$ | $0.74 \pm 0.03$ | $0.69 \pm 0.04$ |
| SimNPO | lr=3e-6, e=3 | $0.59 \pm 0.01$ | $0.48 \pm 0.01$ | $0.54 \pm 0.02$ | $0.90 \pm 0.04$ |
| UNDIAL | lr=4e-6, e=10 | $0.60 \pm 0.01$ | $0.57 \pm 0.01$ | $0.43 \pm 0.01$ | $\mathbf{1.07 \pm 0.02}$ |
| LUNAR | lr=0.0001 | $0.02 \pm 0.01$ | $0.01 \pm 0.00$ | $0.38 \pm 0.00$ | $1.00 \pm 0.04$ |
| $\delta$-Unlearning | lr=7e-6 | $0.28 \pm 0.07$ | $0.78 \pm 0.01$ | $0.69 \pm 0.02$ | $0.13 \pm 0.04$ |
| ULD | lr=2e-3 | $0.40 \pm 0.02$ | $0.40 \pm 0.02$ | $0.39 \pm 0.00$ | $0.42 \pm 0.06$ |
| WHP | lr=1e-5, $\alpha$=3.0 | $0.56 \pm 0.01$ | $0.43 \pm 0.02$ | $0.51 \pm 0.02$ | $0.94 \pm 0.05$ |
| GUARD | lr=1e-3, $\delta$=0.3 | $0.02 \pm 0.01$ | $0.01 \pm 0.00$ | $0.38 \pm 0.00$ | $0.86 \pm 0.04$ |
| ECO | lr=1e-5, str=50 | $0.65 \pm 0.02$ | $0.69 \pm 0.03$ | $0.66 \pm 0.02$ | $0.62 \pm 0.05$ |

Note: Agg. is the harmonic mean of Mem., Priv., and Utility. Each of these is itself the harmonic mean of several metrics. See Appendix F of (Dorna et al., 2025) for details on the construction of these metrics. 99% CIs computed via hierarchical bootstrap resampling

*Table 6.* TOFU Cross Tokenizer Results

| Method | Config | Agg. ↑ | Mem. ↑ | Priv. ↑ | Utility ↑ |
|---|---|---|---|---|---|
| OLMo Linear CT-DD | lr=5e-5, $\alpha$=3.0 | 0.48 | 0.39 | 0.45 | 0.67 |
| OLMo Rank CT-DD | lr=1e-5, $k$=1000 | 0.59 | 0.44 | 0.66 | 0.76 |
| Gemma Linear CT-DD | lr=3e-5, $\alpha$=2.4 | 0.59 | 0.66 | 0.52 | 0.62 |
| Gemma Rank CT-DD | lr=1e-5, $k$=1 | 0.44 | 0.54 | 0.38 | 0.41 |
| Qwen Linear CT-DD | lr=3e-5, $\alpha$=2.9 | 0.56 | 0.54 | 0.58 | 0.55 |
| Qwen Rank CT-DD | lr=1e-5, $k$=1000 | 0.50 | 0.35 | 0.53 | 0.79 |

*Table 7.* All TOFU Divergence Decoding Results

| Size | Method | Param | Agg. ↑ | Agg w/o Priv. ↑ | Mem. ↑ | Priv. ↑ | Utility ↑ |
|---|---|---|---|---|---|---|---|
| 1B | Linear | $\alpha$=0.5 | 0.31 ± 0.02 | 0.28 ± 0.02 | 0.16 ± 0.01 | 0.38 ± 0.00 | 1.00 ± 0.04 |
| 1B | Linear | $\alpha$=1.0 | 0.59 ± 0.01 | 0.63 ± 0.02 | 0.46 ± 0.01 | 0.53 ± 0.02 | 1.00 ± 0.04 |
| 1B | Linear | $\alpha$=1.1 | 0.64 ± 0.01 | 0.65 ± 0.02 | 0.48 ± 0.01 | 0.63 ± 0.02 | 1.00 ± 0.04 |
| 1B | Linear | $\alpha$=1.2 | 0.69 ± 0.01 | 0.67 ± 0.02 | 0.51 ± 0.01 | 0.74 ± 0.03 | 1.00 ± 0.04 |
| 1B | Linear | $\alpha$=1.3 | 0.74 ± 0.01 | 0.69 ± 0.02 | 0.53 ± 0.01 | 0.86 ± 0.03 | 1.00 ± 0.04 |
| 1B | Linear | $\alpha$=1.4 | 0.77 ± 0.02 | 0.71 ± 0.02 | 0.55 ± 0.02 | 0.96 ± 0.02 | 1.00 ± 0.04 |
| 1B | Linear | $\alpha$=1.5 | 0.78 ± 0.01 | 0.72 ± 0.02 | 0.56 ± 0.02 | 0.95 ± 0.03 | 1.00 ± 0.04 |
| 1B | Linear | $\alpha$=2.0 | 0.76 ± 0.01 | 0.78 ± 0.02 | 0.65 ± 0.02 | 0.73 ± 0.02 | 0.99 ± 0.04 |
| 1B | Linear | $\alpha$=2.5 | 0.76 ± 0.01 | 0.82 ± 0.02 | 0.71 ± 0.02 | 0.67 ± 0.01 | 0.97 ± 0.05 |
| 1B | Linear | $\alpha$=3.0 | 0.76 ± 0.01 | 0.84 ± 0.02 | 0.76 ± 0.02 | 0.64 ± 0.01 | 0.93 ± 0.05 |
| 1B | Linear | $\alpha$=3.1 | 0.76 ± 0.01 | 0.84 ± 0.02 | 0.77 ± 0.02 | 0.64 ± 0.01 | 0.92 ± 0.04 |
| 1B | Linear | $\alpha$=3.2 | 0.76 ± 0.01 | 0.84 ± 0.02 | 0.78 ± 0.02 | 0.64 ± 0.01 | 0.92 ± 0.04 |
| 1B | Linear | $\alpha$=3.3 | 0.76 ± 0.01 | 0.85 ± 0.02 | 0.79 ± 0.02 | 0.64 ± 0.01 | 0.91 ± 0.04 |
| 1B | Linear | $\alpha$=3.4 | 0.76 ± 0.01 | 0.84 ± 0.02 | 0.80 ± 0.02 | 0.64 ± 0.01 | 0.90 ± 0.04 |
| 1B | Linear | $\alpha$=3.5 | 0.76 ± 0.01 | 0.85 ± 0.02 | 0.80 ± 0.02 | 0.63 ± 0.01 | 0.89 ± 0.04 |
| 1B | Linear | $\alpha$=4.0 | 0.75 ± 0.01 | 0.83 ± 0.02 | 0.83 ± 0.01 | 0.63 ± 0.01 | 0.82 ± 0.04 |
| 3B | Linear | $\alpha$=0.5 | 0.39 ± 0.01 | 0.38 ± 0.02 | 0.24 ± 0.01 | 0.39 ± 0.00 | 1.01 ± 0.04 |
| 3B | Linear | $\alpha$=1.0 | 0.70 ± 0.01 | 0.68 ± 0.02 | 0.52 ± 0.01 | 0.73 ± 0.02 | 1.00 ± 0.04 |
| 3B | Linear | $\alpha$=1.1 | 0.76 ± 0.01 | 0.70 ± 0.02 | 0.54 ± 0.02 | 0.89 ± 0.03 | 1.00 ± 0.04 |
| 3B | Linear | $\alpha$=1.2 | 0.79 ± 0.02 | 0.72 ± 0.02 | 0.56 ± 0.02 | 0.97 ± 0.02 | 1.00 ± 0.04 |
| 3B | Linear | $\alpha$=1.3 | 0.78 ± 0.02 | 0.73 ± 0.02 | 0.58 ± 0.02 | 0.88 ± 0.02 | 1.00 ± 0.05 |
| 3B | Linear | $\alpha$=1.4 | 0.76 ± 0.01 | 0.75 ± 0.02 | 0.60 ± 0.02 | 0.80 ± 0.02 | 1.00 ± 0.05 |
| 3B | Linear | $\alpha$=1.5 | 0.76 ± 0.01 | 0.76 ± 0.02 | 0.62 ± 0.02 | 0.75 ± 0.02 | 0.99 ± 0.04 |
| 3B | Linear | $\alpha$=2.0 | 0.76 ± 0.01 | 0.82 ± 0.02 | 0.70 ± 0.02 | 0.66 ± 0.01 | 0.97 ± 0.04 |
| 3B | Linear | $\alpha$=2.5 | 0.76 ± 0.01 | 0.84 ± 0.02 | 0.76 ± 0.02 | 0.64 ± 0.01 | 0.94 ± 0.04 |
| 3B | Linear | $\alpha$=2.6 | 0.76 ± 0.01 | 0.85 ± 0.02 | 0.77 ± 0.02 | 0.64 ± 0.01 | 0.94 ± 0.04 |
| 3B | Linear | $\alpha$=2.7 | 0.77 ± 0.01 | 0.85 ± 0.02 | 0.78 ± 0.02 | 0.64 ± 0.01 | 0.94 ± 0.04 |
| 3B | Linear | $\alpha$=2.8 | 0.77 ± 0.01 | 0.86 ± 0.02 | 0.79 ± 0.02 | 0.63 ± 0.01 | 0.93 ± 0.04 |
| 3B | Linear | $\alpha$=2.9 | 0.77 ± 0.01 | 0.86 ± 0.02 | 0.80 ± 0.02 | 0.63 ± 0.01 | 0.92 ± 0.04 |
| 3B | Linear | $\alpha$=3.0 | 0.77 ± 0.01 | 0.86 ± 0.02 | 0.81 ± 0.02 | 0.63 ± 0.01 | 0.91 ± 0.04 |
| 3B | Linear | $\alpha$=3.1 | 0.76 ± 0.01 | 0.86 ± 0.02 | 0.82 ± 0.02 | 0.63 ± 0.01 | 0.90 ± 0.04 |
| 3B | Linear | $\alpha$=3.2 | 0.76 ± 0.01 | 0.85 ± 0.02 | 0.83 ± 0.02 | 0.63 ± 0.01 | 0.88 ± 0.04 |
| 3B | Linear | $\alpha$=3.5 | 0.76 ± 0.01 | 0.85 ± 0.02 | 0.85 ± 0.01 | 0.63 ± 0.01 | 0.85 ± 0.04 |
| 3B | Linear | $\alpha$=4.0 | 0.74 ± 0.01 | 0.82 ± 0.02 | 0.87 ± 0.01 | 0.62 ± 0.01 | 0.77 ± 0.03 |
| 1B | Rank | k=1 | 0.58 ± 0.01 | 0.77 ± 0.02 | 0.64 ± 0.02 | 0.38 ± 0.00 | 0.98 ± 0.04 |
| 1B | Rank | k=5 | 0.73 ± 0.01 | 0.85 ± 0.02 | 0.74 ± 0.01 | 0.57 ± 0.02 | 0.98 ± 0.04 |
| 1B | Rank | k=20 | 0.85 ± 0.02 | 0.87 ± 0.02 | 0.80 ± 0.01 | 0.81 ± 0.02 | 0.95 ± 0.05 |
| 1B | Rank | k=50 | 0.75 ± 0.01 | 0.75 ± 0.02 | 0.63 ± 0.02 | 0.75 ± 0.02 | 0.92 ± 0.04 |
| 1B | Rank | k=100 | 0.81 ± 0.02 | 0.86 ± 0.02 | 0.85 ± 0.01 | 0.73 ± 0.01 | 0.87 ± 0.05 |
| 1B | Rank | k=200 | 0.75 ± 0.01 | 0.76 ± 0.02 | 0.67 ± 0.02 | 0.72 ± 0.01 | 0.88 ± 0.04 |
| 1B | Rank | k=500 | 0.74 ± 0.01 | 0.75 ± 0.02 | 0.72 ± 0.02 | 0.72 ± 0.01 | 0.79 ± 0.04 |
| 1B | Rank | k=1000 | 0.72 ± 0.02 | 0.73 ± 0.02 | 0.75 ± 0.02 | 0.72 ± 0.01 | 0.71 ± 0.04 |
| 3B | Rank | k=1 | 0.60 ± 0.01 | 0.84 ± 0.02 | 0.72 ± 0.02 | 0.38 ± 0.00 | 0.99 ± 0.04 |
| 3B | Rank | k=5 | 0.81 ± 0.01 | 0.89 ± 0.02 | 0.82 ± 0.01 | 0.69 ± 0.02 | 0.97 ± 0.04 |
| 3B | Rank | k=20 | 0.85 ± 0.01 | 0.89 ± 0.02 | 0.86 ± 0.01 | 0.77 ± 0.02 | 0.93 ± 0.04 |
| 3B | Rank | k=50 | 0.76 ± 0.01 | 0.77 ± 0.02 | 0.67 ± 0.02 | 0.74 ± 0.01 | 0.90 ± 0.04 |
| 3B | Rank | k=100 | 0.81 ± 0.02 | 0.85 ± 0.03 | 0.89 ± 0.01 | 0.73 ± 0.01 | 0.82 ± 0.06 |
| 3B | Rank | k=200 | 0.76 ± 0.01 | 0.78 ± 0.02 | 0.73 ± 0.02 | 0.73 ± 0.01 | 0.84 ± 0.04 |
| 3B | Rank | k=500 | 0.76 ± 0.01 | 0.77 ± 0.02 | 0.77 ± 0.02 | 0.72 ± 0.01 | 0.78 ± 0.04 |
| 3B | Rank | k=1000 | 0.74 ± 0.02 | 0.74 ± 0.02 | 0.78 ± 0.02 | 0.72 ± 0.01 | 0.70 ± 0.04 |

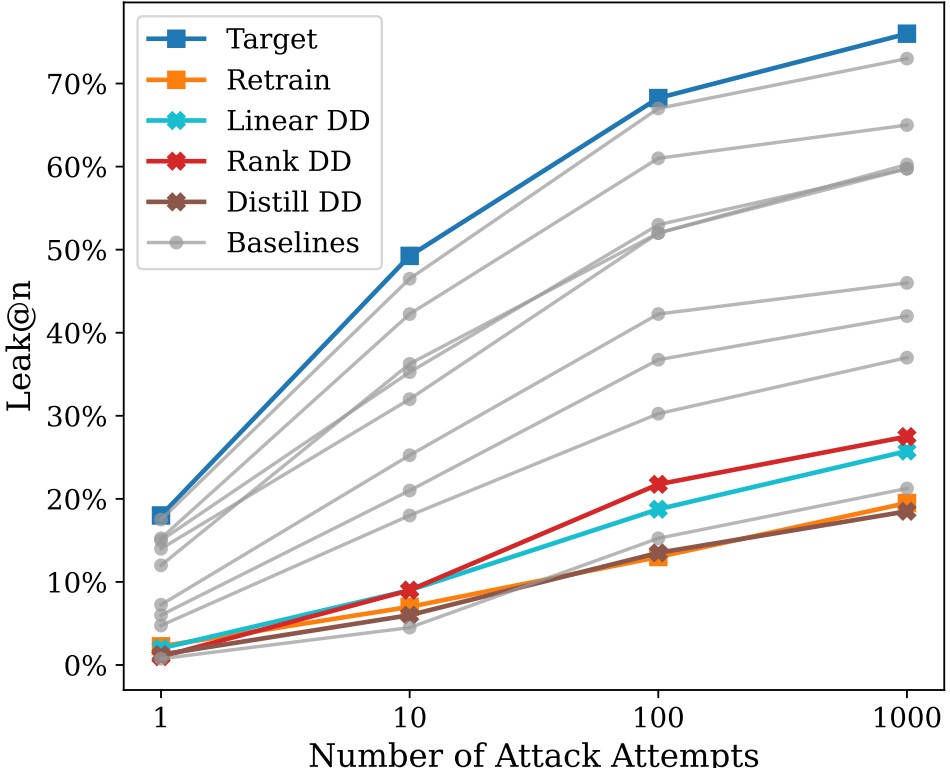

*Figure 17.* Results for Leak@K. We use the code adapted from (Rybak et al., 2026)

# D. Application to Image Generation

In this section we detail the experimental setup used to assess the quality of generated images and additionally present (i) distributional statistics of image quality generated using our divergence decoding setup and (ii) a random sample of generated images for qualitative analysis.

## D.1. Experimental Setup

Each image in our sample is generated using class conditional generation using the default generation parameters of (Esser et al., 2021) for their ImageNet checkpoint. We fine-tune the parameters of auto-regressive transformer in this model to arrive at checkpoints for $p$ and $q$ using a peak learning rate of 10% of that used in (Esser et al., 2021) training for 10 epochs each over the retain and forget sets. We then generate image samples following Eq. 3 where the adjustment from $p$ and $q$ is based solely on the output from the auto-regressive transformer.

## D.2. Measuring Image Content and Quality

Image content is measured using VQAScore (Lin et al., 2024). This approach requires access to the log probabilities of the multi-modal LLM (MLLM) used to assess quality, therefore these assessments rely on the GPT-4o-mini rather than newer models such as GPT-5-nano for which the log probabilities are not exposed. When measuring perceptual quality, we use a MLLM-as-a-judge in a pairwise comparison configuration (Chen et al., 2024). Since this setup does not require access to the log probabilities, we leverage several state of the art small MLLMs for this task: Gemini 2.5 Flash-Lite, GPT-5-nano, and Qwen3-VL 8B.

## D.3. Distributional Properties of Generated Images

Ideally, samples from the model would no longer exhibit image semantics associated with the data in the forget set $F$, while retaining high perceptual quality relative to the retain set $R$. Following prior work (e.g., Heusel et al., 2017), we measure the

*Table 8.* Perceptual quality analysis of images generated using various divergence decoding setups and MLLM judges. Mean values and standard errors are presented.

| Method | Config | Gemini 2.5 Flash-Lite | | GPT-5-nano | | Qwen3-VL 8B | |
|---|---|---|---|---|---|---|---|
| | | Retain | Forget | Retain | Forget | Retain | Forget |
| Linear | $\alpha = 1$ | $52\% \pm 1.7$ | $38\% \pm 1.4$ | $50\% \pm 1.6$ | $38\% \pm 1.4$ | $50\% \pm 1.6$ | $36\% \pm 1.3$ |
| Linear | $\alpha = 2$ | $49\% \pm 1.6$ | $37\% \pm 1.3$ | $49\% \pm 1.6$ | $38\% \pm 1.4$ | $49\% \pm 1.6$ | $39\% \pm 1.4$ |
| Linear | $\alpha = 5$ | $52\% \pm 1.6$ | $31\% \pm 1.2$ | $49\% \pm 1.6$ | $31\% \pm 1.3$ | $52\% \pm 1.6$ | $31\% \pm 1.3$ |
| Linear | $\alpha = 10$ | $47\% \pm 1.6$ | $31\% \pm 1.3$ | $47\% \pm 1.6$ | $32\% \pm 1.3$ | $50\% \pm 1.6$ | $32\% \pm 1.2$ |
| Rank | $topk = 20$ | $49\% \pm 1.6$ | $47\% \pm 1.6$ | $48\% \pm 1.6$ | $47\% \pm 1.6$ | $48\% \pm 1.6$ | $48\% \pm 1.6$ |
| Rank | $topk = 100$ | $47\% \pm 1.6$ | $50\% \pm 1.5$ | $45\% \pm 1.6$ | $46\% \pm 1.5$ | $45\% \pm 1.6$ | $45\% \pm 1.5$ |
| Rank | $topk = 250$ | $48\% \pm 1.6$ | $20\% \pm 1.1$ | $49\% \pm 1.6$ | $21\% \pm 1.1$ | $46\% \pm 1.6$ | $21\% \pm 1.1$ |

quality of the generated images using the Fréchet Inception Distance (FID).

We assess performance by computing the FID between three pairs of data: (i) baseline images from the retain set and generated images using classes from the retain set (FID($B_R,G_R$)), (ii) baseline images from the forget set and generated images using classes from the forget set (FID($B_F,G_F$)), and (iii) baseline images from the retain set and generated images using classes from the forget set (FID($B_R,G_F$)).

Efficacy in this setting preserves perceptual quality relative to the retain set, i.e., low FID($B_R,G_R$) and low FID($B_R,G_F$), while increasing the distance between the forget set and images generated based on those classes, i.e., high FID($B_F,G_F$). In Table 9, we present FID statistics for a variety of decoding setups. For the linear setup, an $\alpha = 1$ seems to work well, e.g., a roughly 33% increase in FID($B_F,G_F$) with only a 5% increase in FID($B_R,G_R$) relative to the baseline. In contrast, the topk based methods appear to require much larger values of $k$ to be effective.

*Table 9.* Content analysis of images generated using various divergence decoding setups.

| Method | Config | FID($B_R,G_R$) $\downarrow$ | FID($B_F,G_F$) $\uparrow$ | FID($B_R,G_F$) $\downarrow$ |
|---|---|---|---|---|
| Baseline | — | 18.2 | 18.0 | 30.1 |
| Linear | $\alpha = 1$ | 19.2 | 24.1 | 27.3 |
| Linear | $\alpha = 2$ | 20.5 | 28.7 | 26.8 |
| Linear | $\alpha = 5$ | 22.8 | 31.6 | 25.8 |
| Linear | $\alpha = 10$ | 22.6 | 31.4 | 25.3 |
| Rank | topk=20 | 19.1 | 20.0 | 29.2 |
| Rank | topk=100 | 20.1 | 22.1 | 28.7 |
| Rank | topk=250 | 21.1 | 28.1 | 26.0 |

## D.4. Image Generation Samples

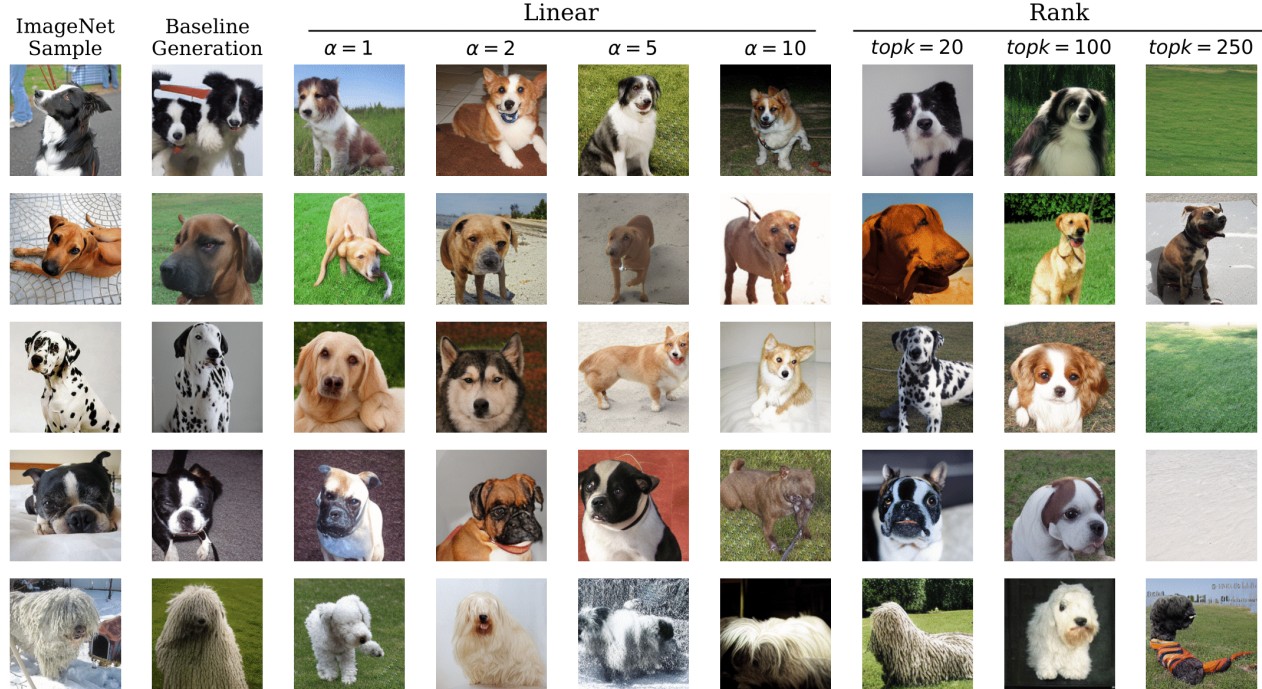

*Figure 18.* Random sample of image generations for classes in the forget set.

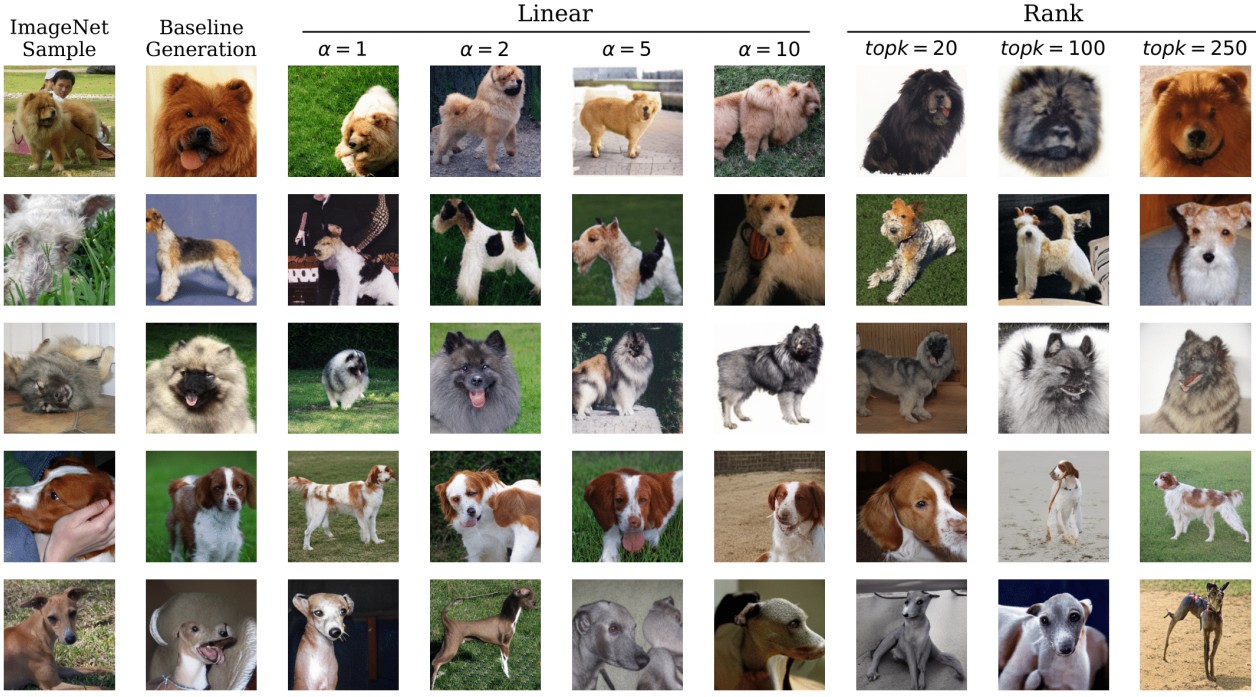

*Figure 19.* Random sample of image generations for classes in the retain set.

