# OpenReview forum: "Divergence Decoding: Inference-Time Unlearning via Auxiliary Models"
_ICML.cc/2026/Conference — ICML 2026 regular_

### Official Review · Reviewer_quFT · 2026-03-09

**Soundness:** 3
**Presentation:** 1
**Significance:** 2
**Originality:** 2
**Overall Recommendation:** 2
**Confidence:** 4

**Summary:**

This paper proposes a divergence-based decoding framework for targeted forgetting, built on the core insight that learning is easier than forgetting. The approach trains two small auxiliary models separately adapted to the forget set and the retain set, and leverages the divergence between their outputs to adjust the logits of a large language model during inference, thereby steering the model away from undesired content. The guiding behavior can further be distilled into the model’s weights, allowing the forgetting capability to be internalized without auxiliary models at inference time.

Experimental results on the TOFU and MUSE benchmarks show that the proposed method significantly outperforms existing approaches, while introducing minimal additional latency and computational overhead. The method effectively balances over-forgetting and under-forgetting, and demonstrates strong scalability and sustainability. Moreover, it successfully generalizes to the image generation domain, where it enables the suppression of specific visual concepts.

**Compliance With Llm Reviewing Policy:**

Affirmed.

**Final Justification:**

Several of my main concerns remain unresolved, and therefore I will keep my original ratings.

**Key Questions For Authors:**

1. What are the dataset contents and the specific evaluation metrics used in the two unlearning benchmarks, MUSE and TOFU?
2. What are the basic principles of the gradient-based baseline methods used in the paper, such as GradDiff, NPO, and SimNPO, and how are these methods implemented in practice?
3. The training of the auxiliary models (p/q) in the paper relies on a strict separation between the forget set and the retain set defined by the benchmarks. In real-world unlearning scenarios, however, the requirements are often unstructured or ambiguous (e.g., removing a class of vaguely defined sensitive information without a clearly labeled forgetting dataset). In such cases, how can the training data distributions for p and q be constructed?
4. What is the specific implementation and role of the trigram language model based on the simple backoff method mentioned in the paper?

**Limitations:**

yes.

**Strengths And Weaknesses:**

Strengths

1. The paper proposes a divergence-based decoding framework guided by small auxiliary models, which avoids retraining the large model from scratch. This effectively addresses the high computational cost of unlearning in frontier models, while introducing only minimal increases in latency and computational overhead.
2. The method significantly outperforms current state-of-the-art approaches on the two authoritative benchmarks, TOFU and MUSE. In particular, on TOFU, which is highly sensitive to fine-tuning, the method achieves performance close to ideal retraining, while effectively balancing over-forgetting and under-forgetting. It also demonstrates strong scalability, sustainability, and robustness.
3. The approach is not limited to textual unlearning in large language models, but is also successfully adapted to the image generation domain in computer vision, where it can effectively suppress specific visual concepts. This demonstrates cross-modal applicability. Moreover, the method shows strong robustness to auxiliary model size and hyperparameters, and even small trigram language models can achieve efficient token-level forgetting.

Weaknesses

1. Although the experimental results on the MUSE benchmark significantly outperform the GradDiff method, they do not appear to show a clear advantage over NPO. In particular, the performance of the distilled model still leaves room for improvement.
2. The paper attempts to explain the relationship between Divergence Decoding and Product of Experts (PoE). However, PoE primarily focuses on the fusion and normalization of multiple expert distributions, whereas Divergence Decoding more closely resembles modifying the sampling distribution of the target model using importance sampling weights derived from auxiliary models. Similar decoding strategies have already been used in safety-oriented contrastive decoding methods. Furthermore, distilling from the probability distribution produced by Divergence Decoding is closer to a form of on-policy distillation. Therefore, the novelty of the proposed method appears somewhat limited.
3. The experimental section contains a large amount of content, but many basic experimental settings are not clearly described. The paper does not provide a detailed introduction of the main baseline methods, and the related literature section also lacks detailed descriptions of these baselines. In addition, the paper does not provide in-depth analysis of experimental and ablation results, and the overall organization and structure of the paper are quite unclear.

---

> ### Author Rebuttal · Authors · 2026-03-31
>
> # Clarifying benchmarks, baselines, and setup (W1, W2, W3, Q1, Q2, Q3, Q4)
>
> We agree that the submission did not describe several benchmark, baseline, and implementation details clearly enough. In the revision, we will make this material more explicit and better organized. Our experiments follow the standard MUSE and TOFU benchmark protocols rather than introducing custom evaluation pipelines.
>
> ## MUSE and TOFU datasets and metrics
>
> Both MUSE and TOFU define an unlearning setting with **forget**, **retain**, and **holdout**-style evaluation splits. A **target** model is trained on both the forget and retain data, while a **retrain** model is trained only on the retain data. The goal is to make the target model behave as similarly as possible to retrain without retraining the large model from scratch.
>
> For **MUSE**, the benchmark includes verbatim memorization on the forget set, knowledge memorization on both forget and retain data, privacy-oriented evaluations based on model behavior on forget versus retain data, and scaling / sustainability evaluations. We consider these metrics jointly, since forgetting and utility come at a tradeoff.
>
> For **TOFU**, the setup is similar, but the benchmark uses instruction-tuned models and question-answer evaluation. In our paper, we follow recent work that reports an aggregate score summarizing the utility–unlearning tradeoff across multiple sub-metrics.
>
> ## Baselines
>
> At a high level, methods such as GradDiff, NPO, and SimNPO apply **opposing updates** (training with negative LR) on forget and retain information. They combine an update that pushes the model away from the forget data, often via gradient ascent or an equivalent preference-style objective, with a retain-side objective that attempts to preserve general utility. In general, these gradient based methods tend to suffer from **catastrophic forgetting** since these objectives are unstable.
>
> ## MUSE comparison to NPO and distilled performance
>
> On **MUSE verbatim memorization**, our methods and the strongest baselines perform similarly. However, on **MUSE knowledge memorization** and **TOFU**, DD variants substantially outperform the main gradient-based baselines, while the distilled model performs in between the baselines and the DD variants despite collapsing the controller into a single model.
>
> We view distillation primarily as evidence that the behavior induced by DD can be transferred back into the model’s weights. The upper bound remains the full DD system.
>
> ## Novelty and relation to prior decoding frameworks
>
> We agree that DD is naturally connected to Product of Experts, Importance Sampling, and related decoding-based methods. Our claim is not that we introduce these underlying ideas. Rather, the contribution is showing that they can be combined into a simple and effective **unlearning system**: two small auxiliary models are trained with ordinary fine-tuning, their divergence is used to steer a large model at inference time, and that behavior can then be distilled back into the model weights.
>
> We also agree that there are parallels to contrastive decoding methods. We will revise the framing to make this relationship clearer and avoid overstating novelty at the level of the underlying mathematical ingredients. The novelty lies in the unlearning formulation, the empirical performance, and the bridge from inference-time steering to weight-space distillation.
>
> ## Constructing p and q beyond benchmark splits
>
> Benchmark unlearning is cleaner than many real-world requests. In our experiments, we train the auxiliary models on the benchmark-provided splits for comparability with prior work.
>
> More generally, the intended practical use case is not that p and q must be trained on two fully disjoint corpora. In a realistic setting, they would typically be trained on **largely overlapping distributions**, where the forget content is a relatively small perturbation to an otherwise shared corpus. In that setting, the divergence between the two small models isolates the targeted information to be suppressed while leaving the rest of the model behavior largely unchanged.
>
> ## Trigram language model
>
> The trigram model is included as an extremely lightweight auxiliary model to show that DD does not strictly require transformer-based experts. Operationally, we precompute next-token probability distributions keyed by short context strings, using unigram, bigram, and trigram counts with simple backoff for unseen contexts. At inference time, these distributions can be combined in the same add/subtract spirit as our logit-space DD controller, but at negligible training and inference cost.
>
> Its role in the paper is to demonstrate that very small and simple auxiliary models can still provide useful forgetting signals.

---

> > ### Author Rebuttal · Reviewer_quFT · 2026-04-04
> >
> > Thanks the responses from the authors. However, some of my concerns are still unaddressed, I will keep my original ratings.

---

> > > ### Author Response · Authors · 2026-04-04
> > >
> > > We appreciate you taking the time to read the rebuttal. If you can highlight the specific concerns from your original review which went unaddressed in the rebuttal, we are happy to provide further clarifications and/or empirical analyses.

---

### Official Review · Reviewer_jmu8 · 2026-03-11

**Soundness:** 3
**Presentation:** 2
**Significance:** 3
**Originality:** 3
**Overall Recommendation:** 4
**Confidence:** 4

**Summary:**

The paper introduces Divergence Decoding (DD), a novel inference-time approach to machine unlearning for Large Language Models (LLMs). Rather than directly updating the weights of a target frontier model to induce forgetting—which is often computationally expensive and prone to catastrophic forgetting—the authors propose steering the target model's logits during generation. This steering is guided by the difference between two small, efficiently trained auxiliary models: one trained on the "forget" set and one on the "retain" set. The authors propose linear and rank-based logit adjustments and demonstrate state-of-the-art performance on the TOFU and MUSE benchmarks. Furthermore, they show that this inference-time behavior can be distilled back into the target model's weights and demonstrate the method's generalizability by applying it to image generation.

**Compliance With Llm Reviewing Policy:**

Affirmed.

**Final Justification:**

My concerns have been addressed in the rebuttal; I maintain my positive rating.

**Key Questions For Authors:**

1. **VRAM Quantification:** Please include a brief analysis or table detailing the peak memory (VRAM) requirements of Divergence Decoding compared to the baseline target model. How does the memory scale when using 1B or 3B auxiliary models alongside an 8B or 70B parameter target model?
2. **Sequential Distillation:** Figure 10 shows that DD is sustainable for sequential unlearning requests at inference time. However, how does the *distillation* approach handle sequential requests? Does the model suffer from catastrophic forgetting if it undergoes multiple sequential distillation phases for different forget sets?
3. **Trigram Model Discrepancy:** The finding that \(n\)-gram models work well on MUSE but fail entirely on TOFU is fascinating. The paper briefly attributes this to MUSE testing "verbatim" memorization. It would strengthen the paper to expand on this: is there a theoretical threshold of task complexity where non-neural auxiliary models break down?

**Limitations:**

yes

**Strengths And Weaknesses:**

**Strengths:**
1. **Conceptual Novelty:** Framing unlearning as an inference-time steering problem using a Product of Experts (PoE) approximation is elegant. It bypasses the fragility of gradient-based weight updates (like gradient ascent) while maintaining high utility.
2. **Strong Empirical Results:** The method achieves near-perfect performance on the TOFU benchmark, which is notoriously difficult due to its use of instruction-tuned models. Outperforming established baselines like NPO, GradDiff, and RMU across both MUSE and TOFU is a significant achievement.
3. **Thorough Ablations and Systems Analysis:** The authors do not shy away from the practical implications of their method. The inclusion of an extensive compute and latency analysis (over 1,200 model combinations) in Section 4.6 and Appendix B is highly commendable and relevant for real-world deployment.
4. **Cross-Domain Validation:** Applying the method to image generation demonstrates that the underlying mathematical formulation is domain-agnostic, which significantly broadens the impact of the work.

**Weaknesses:**
1. **Memory Overhead:** While the compute (FLOPs) and latency overhead are well-analyzed, the paper lacks a detailed discussion on the VRAM/memory overhead. Running three models simultaneously (the target model \(P\) and auxiliary models \(p\) and \(q\)) requires hosting all three sets of weights in memory. For memory-constrained deployments, this could be a dealbreaker.
2. **Distillation Trade-offs:** The proposed solution to inference overhead is standard student-teacher distillation. However, as noted in Appendix B.3, this requires a massive upfront compute investment. If a provider receives frequent, ad-hoc "right to be forgotten" requests, distilling a new model every time becomes computationally prohibitive, limiting the method to either permanent inference-time steering or batch-processed unlearning.
3. **Image Quality Degradation:** In the visual domain, strong unlearning (e.g., \(\alpha = 5\)) successfully suppresses the forget class but severely degrades perceptual quality (only preferred 31% of the time vs. the baseline). This suggests the method might struggle with continuous/latent spaces compared to discrete token distributions.
4. **Dependence on Clean Data Splits:** The methodology assumes access to perfectly cleanly separated "retain" and "forget" datasets to train the auxiliary models. In real-world scenarios, identifying the exact boundaries of a copyrighted text or a sensitive concept without overlapping into safe, general knowledge is highly non-trivial.

---

> ### Author Rebuttal · Authors · 2026-03-31
>
> We appreciate the positive review and the very concrete suggestions. We agree these points would strengthen the paper, and we have clarified them in the revision.
>
> # Memory overhead and VRAM scaling (W1 and Q1)
>
> Thank you for raising this. To quantify the memory overhead of DD, we benchmarked four GPU configurations and measured both raw weight memory (bf16) and the maximum supported batch size at 2048 tokens. The batch-size benchmark captures **both model weights and KV-cache memory**: we determine the maximum batch size by binary search under full 2048-token KV-cache pressure (prefill + decode), so it reflects the true peak VRAM footprint of running all three models together.
>
> **Weight memory (bf16):**
>
> | Config | Weights |
> |---|---:|
> | 8B alone | 14.96 GB |
> | 8B + 2×1B | 19.56 GB |
> | 8B + 2×3B | 26.93 GB |
> | 70B alone | 131.42 GB |
> | 70B + 2×1B | 136.02 GB |
> | 70B + 2×3B | 143.38 GB |
>
> **Max batch size at 2048 tokens (bf16):**
>
> | Config | A40 (44 GB) | H100 (79 GB) | 2×H100 (158 GB) | B200 (178 GB) |
> |---|---:|---:|---:|---:|
> | 8B alone | 30 | 70 | — | 190 |
> | 8B + 2×1B | 15 | 35 | — | 100 |
> | 8B + 2×3B | 5 | 20 | — | 75 |
> | 70B alone | — | — | 12 | 30 |
> | 70B + 2×1B | — | — | 6 | 20 |
> | 70B + 2×3B | — | — | 4 | 15 |
>
> So DD does increase peak VRAM, but reduced max batch size does **not** necessarily imply a proportional throughput loss, since many production setups are already throughput-saturated before reaching the maximum batch size.
>
> # Distillation under sequential requests (W2 and Q2)
>
> We agree that distillation is best viewed as a **batch-processing** solution rather than the ideal mechanism for highly frequent ad hoc requests. More broadly, this tradeoff is not unique to DD: in practice, unlearning methods generally incur either **persistent inference-time overhead** or **repeated gradient updates** when new forget requests arrive. DD makes this tradeoff especially explicit because the same inference-time mechanism can be used directly, or amortized later through distillation.
>
> To address the sequential setting directly, we ran sequential distillation experiments on the MUSE scaling and sustainability protocols and measured the extra distance from raw DD (0 = identical to DD; negative = even closer to retrain).
>
> **Scaling (increasing forget data):**
>
> | Step | Verbmem | Knowmem |
> |---|---:|---:|
> | 1 | +9.3% | +8.9% |
> | 2 | +11.6% | +10.0% |
> | 3 | +18.6% | +23.2% |
> | 4 | +8.6% | +15.1% |
>
> **Sustainability (accumulated forget data):**
>
> | Step | Verbmem | Knowmem |
> |---|---:|---:|
> | 1 | +9.3% | +8.9% |
> | 2 | -4.2% | -15.4% |
> | 3 | +6.2% | +4.9% |
> | 4 | +3.6% | +4.5% |
>
> We therefore do **not** observe evidence of compounding catastrophic degradation under sequential distillation.
>
> # Image quality degradation (W3)
>
> We appreciate this observation. We agree that **very aggressive** forgetting in the image setting can reduce perceptual quality on samples conditioned on the forget class. At the same time, we do not view the $\alpha=5$ result as the practical operating point: in our experiments, **modest** settings already achieve strong forgetting. For example, at $\alpha=1$, the mean VQAScore on forget classes drops from **97% to 20%**, while retain-side FID changes only from **18.2 to 19.2**. We therefore view the 31% preference at $\alpha=5$ as an extreme point on the utility forgetting frontier rather than evidence that DD fundamentally struggles in continuous spaces.
>
> # Dependence on clean forget/retain splits (W4)
>
> We agree that perfectly clean retain/forget boundaries are difficult in real deployments. However, this assumption is shared by benchmarked unlearning methods and is not specific to DD. In practice, if one can define an approximate forget corpus or concept well enough to train a small auxiliary forget model, DD may be especially attractive since adapting the small auxiliaries is much cheaper and less fragile than repeatedly modifying the frontier model itself. We will clarify this limitation explicitly in the revision.
>
> # N-Gram models (Q3)
>
> This is an interesting point. Our current intuition is that n-gram auxiliaries help when forgetting is dominated by **local lexical patterns** (e.g., near-verbatim spans or rare phrase continuations), but break down once the benchmark requires broader instruction following or semantic abstraction. This is consistent with what we observe: MUSE is much closer to local memorization behavior, while TOFU aggregates more complex behaviors on instruction-tuned models. We will add this intuition in the revision.

---

> > ### Author Rebuttal · Reviewer_jmu8 · 2026-04-01
> >
> > I think his response answered my question well. The previous rating is well-justified, in my opinion.

---

> > > ### Author Response · Authors · 2026-04-07
> > >
> > > Thank you for the positive rating! We will integrate the two new experiments into the camera ready. Please let us know if anything else comes to mind.

---

### Official Review · Reviewer_4CMr · 2026-03-12

**Soundness:** 2
**Presentation:** 3
**Significance:** 2
**Originality:** 2
**Overall Recommendation:** 4
**Confidence:** 4

**Summary:**

The paper introduces Divergence Decoding, an inference-time unlearning framework designed to selectively remove sensitive knowledge from LLMs without the computational burden of retraining from scratch or the utility degradation common in gradient-based parameter updates. The method utilizes two small auxiliary models—one fine-tuned on a forget dataset and another on a retain dataset. During text generation, the difference between the logits of these two auxiliary models is used to adjust the pre-softmax logits of the target base model, effectively steering it away from generating the targeted sensitive content. The authors present both a linear and a rank-based logit adjustment method. Finally, to circumvent the inference-time latency of running three models simultaneously, the steered behavioral distribution is distilled back into the base model's weights using a student-teacher KL-divergence objective. The framework demonstrates state-of-the-art utility preservation and forget quality on the TOFU and MUSE benchmarks, alongside evidence of cross-modal applicability in image generation.

**Compliance With Llm Reviewing Policy:**

Affirmed.

**Final Justification:**

The authors have addressed all my concerns.

**Key Questions For Authors:**

1. Could the authors compare the proposed method with the methods mentioned in the weaknesses?
2. Prior to the distillation phase, the purely steered Divergence Decoding model masks behavior without altering the base parameters. Have you tested this intermediate steered model against adversarial prompt injection or deep paraphrasing attacks to evaluate the limits of its behavioral suppression?

**Limitations:**

No. The paper briefly mentions in the ethics statement that the approach could be misused to induce biases, and notes benchmark constraints regarding the retain set, but it lacks a dedicated, rigorous limitations section.

**Strengths And Weaknesses:**

Strengths:
1. The method achieves near-perfect utility preservation on complex, instruction-tuned benchmarks like TOFU, significantly outperforming traditional weight-updating baselines such as NPO and GradDiff.
2. By shifting the unlearning burden to small auxiliary models, DD avoids expensive full-model optimization. The paper provides an exceptionally thorough latency and compute ablation across over 1,200 model configurations, proving the inference overhead is minimal in distributed settings.
3. The inclusion of a final student-teacher distillation phase is a major strength. It elegantly translates the inference-time steering mechanism into a permanent, parametric model update, eliminating deployment latency.
4. Validating the logit-steering approach on VQ-GAN for image generation provides compelling evidence of the method's generalizability beyond text.

Weaknesses:
1. The paper claims state-of-the-art performance but completely omits critical, contemporaneous logit-steering competitors. Methods such as Unlearning via Contrastive Decoding (UCD), Offset Unlearning ($\delta$-Unlearning), and Unlearning from Logit Difference (ULD) employ nearly identical auxiliary-model logit adjustments and have already published highly competitive results on the exact same TOFU and MUSE benchmarks.
2. The authors frame the inference-time logit adjustment as a novel derivation of the Product of Experts framework. However, the exact mathematical mechanism of offsetting a base model with the logit difference of a tuned expert and untuned anti-expert was established by earlier works like DExperts and Proxy-Tuning, which are absent from the literature review.
3. The method's strict requirement for a well-defined retain dataset to train the $q$ model represents a significant logistical bottleneck in real-world scenarios, a limitation that recent forget-data-only methods (like FLAT) have already bypassed.

---

> ### Author Rebuttal · Authors · 2026-03-31
>
> # Positioning relative to prior logit-steering work (W1, W2, and Q1)
>
> Thank you for raising these points. We agree that the original paper did not position the method clearly enough relative to related logit-steering work, and we appreciate the opportunity to correct this.
>
> More specifically, the **linear** form of DD is closely related to prior logit-space steering methods such as DExperts and Proxy-Tuning, as well as contemporaneous unlearning methods such as UCD. We will revise the paper to make this explicit and narrow the novelty claim accordingly. Our intended contribution is therefore **not** the linear logit offset by itself, but the broader framework around it:
> - a rank-based variant,
> - distillation into a single model,
> - a more comprehensive utility/unlearning analysis across TOFU and MUSE,
> - extensive hyper-parameter sweeps on both benchmarks,
> - a detailed compute/latency study, and
> - a cross-modal demonstration.
>
> Following your suggestion, we added **Offset Unlearning**, **ULD**, and **WHP** to our baselines, and swept hyper-parameters for these methods on both MUSE and TOFU. The revised empirical picture is more nuanced than our original framing, and we will update the paper accordingly:
> - on **MUSE verbatim memorization**, $\delta$-Unlearning is the strongest method,
> - on **MUSE knowledge memorization**, DD is the strongest method, but only by a small margin,
> - on **TOFU,** DD is the strongest overall method, with the newly added logit-based baselines being weaker, and
> - **Distill DD** remains a strong single-model deployment version of the method.
>
> # [(CLICK) Link to MUSE Scatter Plot](https://anonymous.4open.science/r/targeted_unlearning_icml2026/OpenUnlearningDD/muse_scatter_plot.png)
>
> | Method | TOFU Agg ↑ | Mem ↑ | Priv ↑ | Util ↑ |
> | --- | ---: | ---: | ---: | ---: |
> | **Linear DD** | **0.78** | **0.56** | **0.95** | **1.00** |
> | **Rank DD** | **0.85** | **0.80** | **0.81** | **0.95** |
> | **Distill DD** | **0.73** | **0.51** | **0.89** | **0.95** |
> | $\delta$-Unlearning | 0.43 | 0.70 | 0.59 | 0.26 |
> | ULD | 0.27 | 0.16 | 0.39 | 0.49 |
> | WHP | 0.56 | 0.44 | 0.51 | 0.93 |
>
> # Dependence on clean forget/retain splits (W3)
>
> We agree that perfectly clean retain/forget boundaries are difficult in real deployments. However, this assumption is shared by benchmarked unlearning methods and is not specific to DD. In practice, if one can define an approximate forget corpus or concept well enough to train a small auxiliary forget model, DD may be especially attractive since adapting the small auxiliaries is much cheaper and less fragile than repeatedly modifying the frontier model itself. We will clarify this limitation explicitly in the revision.
>
> # Additional limitations (Limitations)
>
> We also agree that the paper should state its limitations more explicitly. Beyond the benchmark-defined forget/retain setup, an important practical limitation is that the auxiliary models $p$ and $q$ must remain well aligned with the target model $P$. In particular, if $p$ and $q$ are not aligned with $P$'s instruction-tuned behavior, the steering signal can become unstable or degrade generation quality. We will add this point explicitly in the revision.
>
> # Adversarial prompting against the steered model (Q2)
>
> This is an excellent question. To directly test the intermediate steered model prior to distillation, we additionally evaluated DD on **REBEL** [1], a very recent benchmark built directly on top of TOFU that generates many adversarial prompts for each question. We view this only as a **supplementary stress test**, not a primary benchmark. We chose it mainly because it could be integrated directly into our existing TOFU pipeline during the rebuttal period.
>
> We report **Leak@K**, the fraction of questions for which **at least one** of the top-$K$ attacker prompts successfully elicits the forgotten information. In particular, **Leak@1000** means that for each question, the benchmark generates 1000 attacker prompts, and a question counts as leaked if any one of those 1000 prompts succeeds.
>
> # [(CLICK) Plot of Adversarial Prompting Results](https://anonymous.4open.science/r/targeted_unlearning_icml2026/OpenUnlearningDD/rebel_leak_plot_simple.png)
>
> | Method | Leak@10 ↓ | Leak@1000 ↓ |
> | --- | ---: | ---: |
> | **Retrain** | **7.0** | **19.5** |
> | **Linear DD** | **9.0** | **25.8** |
> | **Rank DD** | **9.0** | **27.5** |
> | **Distill DD** | **6.0** | **18.5** |
> | NPO | 21.0 | 42.0 |
> | RMU | 25.2 | 46.0 |
> | Offset | 42.2 | 65.0 |
> | ULD | 46.5 | 73.0 |
>
> On this supplementary benchmark, both **Linear DD** and **Rank DD** are much closer to **Retrain** than the standard weight-editing baselines, and **Distill DD** is essentially at the retrain level. We therefore do not present REBEL as definitive evidence, but rather as an additional check that the steered model is not merely hiding the behavior under easy prompts.
>
> [1] Rybak et al. (2026), REBEL: Hidden Knowledge Recovery via Evolutionary-Based Evaluation Loop, arXiv.

---

> > ### Author Rebuttal · Reviewer_4CMr · 2026-04-03
> >
> > I'd like to thank the authors for their diligent rebuttal. I will increase my score.

---

> > > ### Author Response · Authors · 2026-04-07
> > >
> > > We really appreciate it! We can assure you all of your suggestions will be included in the final paper; please let us know if anything else comes to mind to include in the camera ready.

---

### Official Review · Reviewer_PysB · 2026-03-13

**Soundness:** 1
**Presentation:** 2
**Significance:** 2
**Originality:** 1
**Overall Recommendation:** 4
**Confidence:** 4

**Summary:**

This paper proposes Divergence Decoding (DD), an unlearning method uses small trained auxiliary models to steer the logits of the LLM away from specific data during inference. The proposed method achieves strong performance on TOFU and MUSE benchmarks.

**Compliance With Llm Reviewing Policy:**

Affirmed.

**Final Justification:**

Given the updates to improve the paper and clarify several points, I am inclined to increase my score to 4.

**Key Questions For Authors:**

1. If DD requires training both a forget model and a retain model, why not directly use the retain model (q) as the unlearned model? Given an input query, the retained model could already produce outputs consistent with unlearning. Have the authors evaluated the performance of using the retrained auxiliary model (q) directly as the unlearned model?
2. Could the authors further clarify the novelty of DD in relation to prior work, particularly existing logit-steering and generation-time unlearning approaches?
3. Could the authors evaluate the robustness of the proposed method under adversarial or out-of-distribution queries?
4. Could the authors provide clearer clarification of the notation and conceptual design, especially regarding the roles of D_A, D_B​, and the auxiliary models p and q in both the theoretical formulation and experimental implementation?
5. Could the authors demonstrate the claimed SOTA performance under a more comprehensive and fair (with same data access) baseline setting?

**Limitations:**

The paper includes an impact statement, but could further discuss the limitations.

**Strengths And Weaknesses:**

## Strengths
- The proposed method outperforms the selected baseline on TOFU dataset.
- The paper evaluates the proposed method beyond text domain, and provides preliminary results for image domain.
- The code is provided for better reproducibility.

## Weaknesses
_Soundness&Significance_&_Originality_

1. **Notation and conceptual inconsistency.** The paper introduces two distributions D_A​ and D_B​ in the theoretical formulation (Line 75-77), where D_B​ is assumed to be a sub-distribution of the full training data D_A​, maybe the forget data (inferred from "the support of D_B is contained within D_A"). However, later sections assign conflicting semantic roles to the auxiliary models p and q: they are first described as being trained on D_A and D_B​, respectively (Line 126-127), but are later interpreted as forget (p) and retain (q) models in the experiments (Section 4.0.1). This makes it unclear what distribution ratio the divergence decoding step is actually approximating, and whether the practical implementation aligns with the theoretical objective.

2. Anther concern is about the underlying design assumption. The DD framework assumes that forget knowledge corresponds to a separable sub-distribution of full training data. However, factual knowledge in LLMs is often distributed across representations. It remains unclear how adjusting output distributions alone can effectively address knowledge entanglement.

3. Intuitively, successful unlearning of DD is to push generated tokens toward those produced by a retained auxiliary model. Therefore, why the DD framework achieves stronger unlearning quality than GradDiff, which has the similar idea towards the retain distribution directly,  requires more detailed experimental analysis and discussion.

4. For the TOFU experiments, the auxiliary models are closely related to the retrained model, as they follow a similar training paradigm. In real-world settings, the pretraining process of a target model is typically unknown, making it difficult to obtain retrained or retrain-like auxiliary models. As such, the current setup mainly demonstrates effectiveness in a controlled setting where the primary difference is model scale (8B vs 1B).


5. **Incomplete baselines.**  Several relevant baselines are missing or inconsistently reported:
	- GA-GDR and NPO-GDR are not evaluated, these methods also use retain data as DD.
	- SimNPO results are reported on MUSE but not on TOFU.
	- WHP and GA-GDR achieve competitive utility and forgetting trade-offs in the original MUSE paper.
	- Since DD does not require fine-tuning the original model, comparisons with other generation-time or training-free unlearning methods (e.g., ECO (SOTA on TOFU) [1] or GUARD [2]) would be helpful.
	- In addition, several logit or representation level unlearning methods are closely related to DD. For example, UNDIAL [3] modifies logits via distribution subtraction and then distills a student model; ULD [4] derives an unlearned model through logit differences between the target and an auxiliary model trained on forget data; and LUNAR [5] redirects internal representations toward refusal regions. The differences and novelty of DD relative to these approaches should be more clearly articulated.

The paper claims that DD "decisively outperforms SOTA baselines". However, since only a limited subset of baselines is evaluated, this claim may be overstated. A broader and more comprehensive comparison would be necessary to support such a "SOTA" conclusion.


6. The robustness of the method is not sufficiently evaluated. For instance, since DD relies on auxiliary models trained on specific forget and retain sets, it is unclear whether the method generalizes to paraphrased or distribution-shifted (OOD) queries related to the forget data.



_Presentation_

- In Line 87, citation formatting should follow the ICML template (citet{}).
- According to the ICML style guidelines, table captions should be placed above the table (Table 1 & Table 2).
- When introducing tasks beyond the text domain, the paper should clearly describe the experimental setup with good structure, including the target model, forget and retain sets, evaluation metrics, and baseline methods.


Minor:
In Table 1, it is unclear why Linear DD and Rank DD achieve higher Agg. performance than the Retrain model.

[1] Liu, Chris, et al. "Large language model unlearning via embedding-corrupted prompts." _Advances in Neural Information Processing Systems_ 37 (2024): 118198-118266.

[2] Deng, Zhijie, et al. "Guard: Generation-time llm unlearning via adaptive restriction and detection." _arXiv preprint arXiv:2505.13312_ (2025).

[3] Dong, Yijiang River, et al. "Undial: Self-distillation with adjusted logits for robust unlearning in large language models." _Proceedings of the 2025 Conference of the Nations of the Americas Chapter of the Association for Computational Linguistics: Human Language Technologies (Volume 1: Long Papers)_. 2025.

[4] Ji, Jiabao, et al. "Reversing the forget-retain objectives: An efficient llm unlearning framework from logit difference." _Advances in Neural Information Processing Systems_ 37 (2024): 12581-12611.

[5] Shen, William F., et al. "LLM unlearning via neural activation redirection." _arXiv preprint arXiv:2502.07218_ (2025).

---

> ### Author Rebuttal · Authors · 2026-03-31
>
> # Notation and conceptual consistency (W1, Q4)
>
> We apologize for the inconsistencies and this will all be fixed in the final draft. In our theoretical setup, $D_A$ is our forget+retain/target/$P$ distribution and $D_B$ is our goal retain/retrain/$Q$ distribution.
>
> # Distributed / entangled knowledge (W2)
>
> LLMs are trained with a generative supervised objective in output space, so output-space interventions are a natural place to act. More importantly, the privacy metrics in these benchmarks are explicitly designed to measure whether the forgotten knowledge is still present, and DD performs very strongly on them.
>
> Finally, the benchmark setup itself already reflects entanglement. The forget and retain data are not cleanly separable in a mechanistic sense, which is visible in the fact that even the retrain model does not get close to 0 on all forget-related statistics.
>
> # Why DD outperforms GradDiff (W3)
>
> Our intuition is that the objective optimized by methods such as GradDiff is structurally unstable. Gradient ascent on forget data pushes the model toward increasing a loss that is unbounded above, so even when this is partially offset by retain-side updates, the optimization can be brittle and can damage the surrounding behavior of the model. By contrast, DD is built from standard learning on the small auxiliary models, so the unlearning signal comes from a much more stable objective.
>
> # Why not directly use $q$? (W4, Q1)
>
> The point of DD is to preserve the intelligence of the large model $P$ while moving its behavior toward the desired post-unlearning model $Q$. The small retain model $q$ is only a proxy for the directional correction; it is not meant to replace the large model. Using $q$ directly throws away most of the capability of the large model and reduces the problem to running a much smaller model. To evaluate this directly:
>
> **MUSE**
> | Method | VerbMem (Forget) | KnowMem (Forget) | KnowMem (Retain) |
> |--------|-----------------|-------------------|-------------------|
> | Target (7B) | 57.8 | 64.9 | 53.6 |
> | Retrain (7B) | 20.2 | 33.4 | 55.1 |
> | Target (1.3B) | 18.7 | 13.6 | 22.5 |
> | Retrain (1.3B) | 23.4 | 14.8 | 18.4 |
>
> **TOFU**
> | Method | Agg | Mem | Priv | Util |
> |--------|-----|-----|------|------|
> | Target | 0.02 | 0.01 | 0.38 | 1.00 |
> | Retrain | 0.78 | 0.53 | 0.98 | 1.03 |
> | Target (1B) | 0.20 | 0.09 | 0.38 | 0.97 |
> | Retrain (1B) | 0.78 | 0.55 | 0.98 | 0.98 |
>
> We see very clearly that $q$ alone is not sufficient on MUSE. For TOFU, $q$ is very close to $Q$, which is reasonable given the benchmark’s small dataset.
>
> # Baselines and strength of claim (W5, Q5)
>
> We want to emphasize that nearly all methods presented use both the retain and forget datasets. In addition, we sweep relevant hyper-parameters for all methods to provide a fair comparison.
>
> We expanded the baseline set. GradDiff is equivalent to GA-GDR (Gradient Ascent - Gradient Descent Retain.) We also added Offset Unlearning, ULD, UNDIAL, and WHP. On MUSE knowledge memorization and TOFU, DD is still the strongest overall method.
>
> **MUSE**
>
> # [(CLICK) Link to MUSE Scatter Plot](https://anonymous.4open.science/r/targeted_unlearning_icml2026/OpenUnlearningDD/muse_scatter_plot.png)
>
> **TOFU**
>
> | Method | Config | Agg ↑ | Mem ↑ | Priv ↑ | Util ↑ |
> |---|---|---|---|---|---|
> | **Linear DD** | **$\alpha$=1.5** | **0.78** | **0.56** | **0.95** | **1.00** |
> | UNDIAL | lr=4e-6, e=10 | 0.60 | 0.57 | 0.43 | 1.07 |
> | $\delta$-Unlearning | lr=7e-6 | 0.43 | 0.70 | 0.59 | 0.26 |
> | ULD | lr=2e-3 | 0.27 | 0.16 | 0.39 | 0.49 |
> | WHP | lr=1e-5, $\alpha=3.0$ | 0.56 | 0.44 | 0.51 | 0.93 |
>
> # Relation to prior logit methods (Q2)
>
> The linear form of DD is closely related to contemporaneous unlearning methods. The contribution is not the linear logit offset by itself, but the broader framework around it:
>
> - a rank-based variant,
> - distillation into a single model,
> - a broader utility/unlearning analysis across TOFU and MUSE,
> - extensive hyper-parameter sweeps,
> - a compute/latency study,
> - and a cross-modal demonstration.
>
> # Robustness to adversarial / OOD queries (Q3)
>
> We added a new adversarial prompting evaluation using REBEL [1], which builds directly on TOFU. It generates many attack prompts designed to leak the forgotten information. We report LEAK@ metrics, which measure the percentage of questions for which at least one prompt successfully leaks information.
>
> # [(CLICK) Plot of Adversarial Prompting Results](https://anonymous.4open.science/r/targeted_unlearning_icml2026/OpenUnlearningDD/rebel_leak_plot_simple.png)
>
> DD is close to retrain, and Distill DD matches retrain nearly perfectly on this benchmark. We are aware that REBEL is new and not yet vetted by the community; we include it as a supplementary stress test rather than a definitive benchmark.
>
> [1] Rybak et al. (2026), REBEL: Hidden Knowledge Recovery via Evolutionary-Based Evaluation Loop, arXiv.

---

> > ### Author Rebuttal · Reviewer_PysB · 2026-04-03
> >
> > Thank you for the detailed rebuttal. I appreciate the authors' efforts in addressing the concerns and providing additional experimental results. In particular, the comparison with the retrained auxiliary model helps clarify the role of the retain model, and the additional baselines and adversarial evaluation (REBEL) strengthens the empirical results.
> >
> > However, several concerns remain only partially addressed:
> > 1. Weakness 2 (knowledge entanglement): The response mainly provides empirical evidence, but does not directly address the core conceptual question of whether output-level interventions are sufficient to handle entangled and distributed knowledge representations in LLMs.
> > 2. Weakness 4 (practicality of auxiliary models): The rebuttal mainly focuses on why not directly use the retain model q, but does not address my original concern. In the TOFU experiments, the auxiliary models are closely aligned with the retrained model, as they follow a similar training paradigm. In real-world scenarios, however, the pretraining process of the target model is often unknown, making it difficult to obtain retrained or retrain-like auxiliary models. For example, if the target model is something like GPT-OSS-20B, one may only have access to auxiliary models from a different family (e.g., Qwen), which raises questions about the applicability of the method across different model families.
> > 3. Baselines and related work: While additional baselines have been included, the evaluation is still not fully comprehensive, and several relevant methods mentioned in the review would benefit from more discussion about the difference.
> >
> > Overall, I appreciate the authors' efforts in improving the paper and clarifying several points. Given these updates, I am inclined to increase my score to **3**, as my concerns are partially addressed.

---

> > > ### Author Response · Authors · 2026-04-07
> > >
> > > We apologize for the delay and would like to resolve the rest of your concerns.
> > >
> > > # W2
> > >
> > > We froze the target model and computed the DD distillation loss (output KL), then backpropagated without updating weights. We measured the $L_2$ norm of gradients on the MLP and attention weights at each layer.
> > >
> > > ## [(CLICK) Link to W2 Experiment Results](https://anonymous.4open.science/r/targeted_unlearning_icml2026/OpenUnlearningDD/gradient_depth_combined.png)
> > >
> > > We find that the gradient propagates throughout the network and peaks in intermediate layers where prior work suggests factual associations are often stored [1].
> > >
> > > # W4
> > >
> > > We also agree it is important to test whether DD depends on retrain-like auxiliaries from the same family. To address this, we evaluated cross-family auxiliaries on MUSE (and the instruct variants for TOFU):
> > >
> > > - allenai/OLMo-2-0425-1B
> > > - google/gemma-3-1b-pt
> > > - Qwen/Qwen3-1.7B-Base
> > >
> > > We implemented a simple cross-tokenizer bridge. At initialization, each auxiliary token is decoded to text and re-encoded with the main tokenizer to find exact matches; if no exact match exists, we progressively shorten prefixes until a match is found. For example, if the auxiliary has “abcd” and the main model contains “abcd”, “abcde”, and “abcdef”, we map the auxiliary token’s logit difference onto all compatible main-model tokens. This mapping is built once and then applied at inference with a single vectorized scatter, so the additional overhead is negligible.
> > >
> > > ## [(CLICK) Link to MUSE Cross Tokenizer Results](https://anonymous.4open.science/r/targeted_unlearning_icml2026/OpenUnlearningDD/muse_cross_tok_scatter.png)
> > >
> > > **(TOFU Results Underneath)**
> > >
> > > We find that these cross-family auxiliaries remain close to the same-family baseline and still competitive with the broader set of unlearning methods.
> > >
> > > # Baselines and Related Work
> > >
> > > We added ECO, GUARD, and LUNAR, covering the suggested baselines. We also appreciate the reviewer’s push to clarify the distinction between DD and nearby methods. **We will overhaul the literature review section in the camera ready.**
> > >
> > > **ULD:** subtracts a single forget model at low $\alpha$. It does not use a second retain model to partially cancel the shift, so in our view it is a simpler one-sided logit correction.
> > >
> > > **UNDIAL:** subtracts one-hot logits during distillation. This is a much more aggressive intervention than DD, which uses a softer distributional correction.
> > >
> > > **$\delta$-Unlearning:** This is probably the closest prior method to the linear form of DD, but there is a key difference. This method fine-tunes the models as an ensemble (larger training cost since forward passes on all 3 models), only updating a single model. Empirically, this outperforms on verbatim memorization but underperforms on TOFU and knowledge memorization relative to DD.
> > >
> > > **WHP:** also performs linear steering, but uses full-size models (can be implemented with LoRA) and only a forget model.
> > >
> > > **GUARD / ECO:** use binary detection/filtering style mechanisms to decide when to corrupt or block a prompt. These methods do very well on the simplest settings, especially MUSE Verbatim Memorization.
> > >
> > > All other methods are conceptually more different from the above ones, to the best of our knowledge.
> > >
> > > So we agree that the **linear** form of DD is related to contemporaneous logit-steering methods. Our contribution is therefore the broader DD framework:
> > > - a rank-based steering variant,
> > > - a bridge between inference-time unlearning and weight-based unlearning via distillation,
> > > - a bridge to multimodal domains,
> > > - compute/latency ablations,
> > > - and now cross-tokenizer / cross-family evidence.
> > >
> > > ## Results
> > >
> > > **We sweep hyperparameters on all methods and select the optimal configuration for each benchmark.**
> > >
> > > ## [(CLICK) Link to MUSE Gradient Based Methods ](https://anonymous.4open.science/r/targeted_unlearning_icml2026/OpenUnlearningDD/muse_scatter_gradient.png)
> > >
> > > ## [(CLICK) Link to MUSE Inference Time Methods ](https://anonymous.4open.science/r/targeted_unlearning_icml2026/OpenUnlearningDD/muse_scatter_inference.png)
> > >
> > > ## [(CLICK) Link to full MUSE Results](https://anonymous.4open.science/r/targeted_unlearning_icml2026/OpenUnlearningDD/muse_scatter_plot.png)
> > >
> > > ## [(CLICK) Link to TOFU Table](https://anonymous.4open.science/r/targeted_unlearning_icml2026/OpenUnlearningDD/tofu_table.png)
> > >
> > > # Conclusion
> > >
> > > We appreciate the reviewer’s emphasis on both broader baselines and more careful conceptual justification. This helped us sharpen the paper considerably.
> > >
> > > Our updated picture is:
> > > - DD is strongest on the more complex, instruction-tuned setting of **TOFU**,
> > > - DD is marginally best on **MUSE KnowMem**,
> > > - and simpler filtering/classification-style methods can be stronger on the easiest memorization setting, **MUSE Verbatim**.
> > >
> > > We thank you for your willingness to increase the score and for the detailed feedback and back-and-forth.
> > >
> > > [1] Meng et al. (2023), *Locating and Editing Factual Associations in GPT*, NeurIPS 2022

---

### Decision · Program_Chairs · 2026-04-30

**Decision:**

Accept (regular)

**Comment:**

The paper proposes divergence decoding, an inference time distribution tilting to encourage unlearning. The native reviewers who are engaged in discussion are convinced by the additional empirical results and the authors’ feedback. The authors were active, engaging, and responsive during the discussion period, and were able to convince the reviewers of the novelty and the quality of the work.